# Nutritional Epigenomics: Bioactive Dietary Compounds in the Epigenetic Regulation of Osteoarthritis

**DOI:** 10.3390/ph17091148

**Published:** 2024-08-30

**Authors:** Karla Mariuxi Villagrán-Andrade, Carmen Núñez-Carro, Francisco J. Blanco, María C. de Andrés

**Affiliations:** 1Unidad de Epigenética, Grupo de Investigación en Reumatología (GIR), Instituto de Investigación Biomédica de A Coruña (INIBIC), Complexo Hospitalario Universitario, de A Coruña (CHUAC), Sergas, 15006 A Coruña, Spain; kariuxva@hotmail.com (K.M.V.-A.); carmen.nunez.carro@sergas.es (C.N.-C.); fblagar@sergas.es (F.J.B.); 2Grupo de Investigación en Reumatología y Salud, Departamento de Fisioterapia, Medicina y Ciencias Biomédicas, Facultad de Fisioterapia, Campus de Oza, Universidade da Coruña (UDC), 15008 A Coruña, Spain

**Keywords:** nutriepigenomics, osteoarthritis, chondrocyte, cartilage, bioactive compounds, epigenetics

## Abstract

Nutritional epigenomics is exceptionally important because it describes the complex interactions among food compounds and epigenome modifications. Phytonutrients or bioactive compounds, which are secondary metabolites of plants, can protect against osteoarthritis by suppressing the expression of inflammatory and catabolic mediators, modulating epigenetic changes in DNA methylation, and the histone or chromatin remodelling of key inflammatory genes and noncoding RNAs. The combination of natural epigenetic modulators is crucial because of their additive and synergistic effects, safety and therapeutic efficacy, and lower adverse effects than conventional pharmacology in the treatment of osteoarthritis. In this review, we have summarized the chondroprotective properties of bioactive compounds used for the management, treatment, or prevention of osteoarthritis in both human and animal studies. However, further research is needed into bioactive compounds used as epigenetic modulators in osteoarthritis, in order to determine their potential value for future clinical applications in osteoarthritic patients as well as their relation with the genomic and nutritional environment, in order to personalize food and nutrition together with disease prevention.

## 1. Osteoarthritis, a Chronic Disease

Osteoarthritis (OA) is one of the most common disabling chronic progressive diseases in middle-aged and elderly people [1,2], and it is among the main public health problems worldwide, due to its high prevalence [3]. The main characteristics of OA are articular cartilage deterioration, subchondral bone remodelling, the formation of osteophytes, joint space reduction, and synovitis [4]. Symptoms generally include severe joint pain, stiffness, joint contractures, muscle atrophy, reduced movement, swelling, tenderness, and variable degrees of local inflammation, limb deformity and crepitus [5]. There are many etiological factors for OA, including genetic predisposition, dietary intake, obesity, sex, aging, traumatic joint injury, mechanical stress, metabolic disease, and sedentary lifestyle [6]. It is important to highlight the synergistic effects of pathologies such as cardiovascular disease and obesity coexisting with OA [7,8].

Pharmacological treatments such as paracetamol, nonsteroidal anti-inflammatory drugs, tramadol, and opioids are used to reduce pain and inflammation, but do not prevent, reverse or cure OA [9]. However, a long-term use of these drugs to relieve OA is associated with substantial gastrointestinal, renal, hepatic, blood, cardiovascular, and cerebrovascular adverse effects [10,11,12]. In this review, we present the importance of a healthy diet in preventing the development or progression of OA, and summarize chondroprotective properties and beneficial epigenetic modifications of bioactive compounds or nutraceuticals against inflammation and catabolic activity in OA.

## 2. Epigenetics and Osteoarthritis

Over the last 20 years, the study of epigenetics has expanded (especially in the cancer field). However, studies on the importance of epigenetic mechanisms in OA are only now increasing. Roach and collaborators provided the first evidence of how epigenetic changes, such as DNA methylation, may relate to the pathogenesis of OA and can be potentially reversible [13].

Epigenetics can be defined as heritable changes in gene expression and/or phenotype that can occur without changes in the primary DNA sequence [14]. The epigenome of each cell is unique and can undergo temporal changes in response to environmental factors such as diet, physical activity, smoking, pollutants and disease status [15]. OA is distinguished by the unfavorable dynamic regulation of gene transcription in joint tissues due to environmental disturbances; therefore, epigenetics has developed as a new and important area for OA research [16,17,18]. Candidate gene and epigenome-wide studies have demonstrated their association with OA development and progression through epigenetic modifications, and these epigenetic mechanisms can change in response to stimuli and, in some cases, pass on to future generations [19,20,21]. Given the importance of gene expression or silencing, and associated epigenetic modifications, we will briefly mention various epigenetic mechanisms of pro-inflammatory cytokines and metalloproteinases (MMPs) that contribute to cartilage destruction. Three main mechanisms are implicated in epigenetic regulation: (1) DNA methylation changes that covalently alter chromatin structure. In general, DNA hypomethylation enhances gene transcription, and DNA hypermethylation suppresses gene transcription. (2) The post-translational modification of histones that alters chromatin conformation, including the methylation of arginine and lysine, the acetylation of lysines, the phosphorylation of serine and treonine, and the sumoylation and ubiquitination of lysine. (3) Non-coding RNAs regulate gene expression but do not translate into proteins (i.e., microRNAs (miRNAs), long non-coding RNAs) acting at both transcriptional and post-transcriptional levels [22,23,24].

### 2.1. DNA Methylation

The DNA methylation process is mediated by DNA methyltransferases (DNMTs), including DNMT1 (maintenance), DNMT3A and DNMT3B (*de novo*), and involves the addition of a methyl group to the 5′ position of cytosine, which most commonly occurs in CpG dinucleotides, forming 5-methylcytosine. The hypermethylation by DNMTs leads to transcriptional gene silencing and gene inactivation [22,23].

Nakano and collaborators found that *DNMT1* and *DNMT3A* expressions were decreased by IL-1β, while *DNMT3A* also decreased its expression and activity, caused by the TNF-α in fibroblast-like synoviocytes [25]. Both DNA methylation and histone modification are involved in the control of TNF-α expression [26]. Hashimoto and collaborators found that the methylation of the −115 CpG site enhances *MMP13* promoter activity as opposed to the inhibitory effect of −110 CpG methylation; also, the demethylation of the specific CpG sites at the −299 position of the *IL1B* promoter activity correlates with enhanced *IL1B* gene expression in human primary chondrocytes [27,28]. Furthermore, Bui and collaborators showed that the −104 CpG site is demethylated in OA cartilage, and this is accompanied by elevated *MMP13* expression [29]. In articular cartilage, the methylation of cytosines at positions −1680 and −1674 blocks *COL10A1* expression in chondrocytes, while gene expression is activated during chondrogenesis in cells, with the partial methylation of these two specific CpG sites [30]. Cheung and collaborators found that DNA demethylation at one or more specific CpG sites in the *ADAMTS4* promoter corresponds to the increased expression of *ADAMTS4* in human OA chondrocytes, which plays a role in aggrecan degradation in OA [31]. In addition, Roach and collaborators showed an association between the loss of DNA methylation of CpG sites in the promoters and the abnormal expression of *MMP3*, *MMP9*, *MMP13*, and *ADAMTS4* by OA chondrocytes [13]. Besides this, the sclerotin (*SOST*) mRNA and protein expression levels are increased in OA chondrocytes, suggesting the *SOST* promoter is hypermethylated in normal chondrocytes and hypomethylated in OA [32]. An interesting study suggests that hip OA is associated with reduced *SOX9* gene and protein expression, having showed that that the methylation of the *SOX9* promoter was increased in OA cartilage [33]. Imagawa and collaborators reported that *COL9A1* promoter activity is significantly decreased by DNA hypermethylation, and could be reversed through the inhibition of DNA methylation. In addition, the abnormal DNA methylation of the CpG sites in the *COL9A1* promoter is associated with the decreased expression of *SOX9* [34]. Moreover, hypomethylation in the *IL8* promoter is correlated with higher *IL8* gene expression in OA chondrocytes; a significant increase in *IL8* promoter activity by the transcription factors NF-κB, AP-1 and C/EBP was also shown [35]. de Andrés and collaborators demonstrated the association between an increase in inducible nitric oxide synthase (*NOS2*) gene expression in OA chondrocytes and the demethylation of NF-κB responsive enhancer elements [36]. Furthermore, in OA, synovial fibroblasts showed DNA hypomethylation and histone hyperacetylation in the *IL6* promoter [37].

### 2.2. Histone Modifications

Methylation/demethylation and acetylation/deacetylation are the main and recurrent histone changes in OA [38]. Two families of enzymes catalyze the modification of histones: histone methyltransferases (HMTs) and histone demethylases (HDMTs), or acetyltransferases (HATs) and histone deacetylases (HDACs) [39]. The majority of these modifications take place in lysine, arginine and serine residues within the histone tails, and regulate key cellular processes such as transcription, replication and repair [40]. The hyperacetylation of histone tails induces transcriptional activation, while hypoacetylation is associated with transcriptional repression [41]. HDAC family members have been associated with OA, and HDAC inhibitors (HDACi) can protect chondrocytes and prevent cartilage damage, while possessing therapeutic potential against OA [15,42]. Young and collaborators demonstrated that HDACi decreased the expression and activity of MMPs and ADAMTSs [43]. In addition, histone deacetylase-1 (HDAC1) and HDAC2 levels are elevated in both chondrocytes and synovium from OA patients compared to controls [44,45]. Higashiyama and collaborators demonstrated the increased expression of HDAC7 in human OA cartilage, which was correlated with elevated *MMP13* gene expression, contributing to cartilage degradation [46]. Class III HDACs (sirtuins) are a class of NAD+-dependent histone deacetylases that differ from the class I and II HDACs. Sirtuin 1 (SIRT-1) is a positive regulator of cartilage-specific gene expression in chondrocytes [47]. SIRT-1 activation has the potential to prevent cartilage damage and inhibit its destruction [48,49]. SIRT-1 suppresses protein tyrosine phosphatase 1B and activates the insulin-like growth factor (IGF) receptor pathway, enhancing the survival of chondrocytes [50]. Also, the decreased expression of *COL2A1* mRNA and type II collagen protein correlates with decreased SIRT1 activity [51]. In addition, in OA cartilage, the overexpression of E74-like factor 3 (ELF3) inhibited Sox9/cAMP-response element-binding (CREB) protein (CBP)-driven HAT activity, and decreased *COL2A1* [52]. The disruptor of telomeric silencing, the 1-like (*DOT1L*) gene (an HMT), is a protector of cartilage health, and as such is reduced in damaged areas of OA joints; the protective function of DOT1L is attributable to Wnt signalling inhibition [53,54].

### 2.3. Non-Coding RNA (ncRNAs)

ncRNAs, including small non-coding RNAs (miRNA) and long non-coding RNAs (lncRNAs), have the ability to regulate gene expression at both transcriptional (lncRNAs) and post-transcriptional levels (small and lncRNAs) [55]. lncRNAs are key regulators of gene expression; thus, the overexpression of lncRNA-CIR increases the expression of MMPs, whereas collagen and aggrecan expression are reduced in OA cartilage [56]. Small ncRNA mainly includes miRNAs, siRNAs and piRNAs. miRNAs have historically been the most frequently investigated; they are considered an alternative mechanism of post-transcriptional or translational regulation. At the post-transcriptional level, they bind to complementary mRNA, leading to the degradation of mRNA or the prevention of its translation into a protein [55,57,58,59]. Several miRNAs have shown altered expressions in OA, and are involved in various aspects of cartilage homeostasis and OA pathogenesis [60]. Rasheed and collaborators showed that IL-1β-induced *iNOS* gene expression is correlated with the down-regulation of miR-26a-5p in human OA chondrocytes [61]. Furthermore, miRNAs such as miR-320, miR-381, miR-9, miR-602, miR-608, miR-127-5p, miR-140, miR-27b, miR-98 and miR-146 play a significant role in the regulation of genes relevant to OA pathogenesis [59]. In another study, the overexpression of miR-27b inhibited IL-1β-stimulated *MMP13* gene and protein expression in human OA chondrocytes [62]. Moreover, the overexpression of miR-558 directly inhibited *COX2* mRNA and protein expression [63]. Also, miR-199a levels are inversely correlated with *COX2* mRNA and protein levels in IL-1β-stimulated human chondrocytes [64]. There is a relationship between HDACs and miRNA in OA; thus, the overexpression of miR-92a-3p suppressed HDAC2 production and increased the level of histone H3 acetylation of the *COMP/ACAN/COL2A1* promoter [65]. The overexpression of miR-193b-3p inhibited *HDAC3* expression, enhanced histone H3 hyperacetylation, and increased the expressions of *SOX9*, *COL2A1*, *ACAN*, and *COMP* in chondrocytes [66]. Guan and collaborators showed that miR-146a protects against OA, inhibiting inflammatory factors [67]. In addition, a study demonstrated the significant increase in miR-146a expression that was induced by the HDAC inhibitors in OA-fibroblast-like synoviocytes [68]. Another study demonstrated that miR-146b is downregulated in the chondrogenic differentiation of human stem cells, and upregulated in OA [69]. The overexpression of miR193b-5p inhibited *HDAC7* expression and decreased *MMP3* and *MMP13* expression [70]. Both miR-199a-3p and miR-193b expressions are upregulated with age, and may be involved in chondrocyte senescence by downregulating anabolic factors such as type 2 collagen, aggrecan, and SOX9; therefore, they may be involved in cartilage degeneration [71]. In addition, the increases in *TNFA*, *IL1B* and *IL6* gene expression were correlated with miR-149 downregulation through the inhibition of post-transcriptional control in human OA chondrocytes [72]. miR-140, the most well-studied miRNA in OA, plays a protective role in OA development. It is important for chondrogenesis and osteogenesis, and is highly expressed in normal cartilage, but its expression levels are decreased in OA chondrocytes; its overexpression could inhibit inflammation and cartilage degradation [73,74,75,76,77]. A study showed that miR-140 is specifically expressed in cartilage tissues during mouse embryonic development, and that siRNA-140 significantly downregulated the accumulation of the Hdac4 protein in fibroblast cells [78]. Further, miR-140-3p and its isomiRs (miR-140-3p.1 and miR-140-3p.2) are abundantly expressed in cartilage [79]. Decreased miR-let7e expression has been suggested as a potential predictor of hip OA [57,80]. The increase in miR-145 levels directly represses *SOX9* expression, resulting in the inhibition of *COL2A1* and *ACAN*, with increased expressions of *RUNX2* and *MMP13* in human chondrocytes [81].

## 3. Inflammation and Diet

Inflammation is a complex biological response of the immune system to pathogens, damaged cells, injury, toxic compounds, and infection. The immune system utilizes a large number of specialized cells, such as lymphocyte, monocytes and macrophages, to restore homeostasis [82,83,84]. Inflammation is an important pathway in OA pathogenesis and development [85,86]. Inflammation in OA joints is chronic and low-grade, and involves the interplay of the innate immune system and inflammatory mediators [85,87,88]. These include cytokines, chemokines, growth factors, adipokines, prostaglandins, leukotrienes, nitric oxide, and neuropeptides [87,89]. Strikingly, reductions in this low-grade inflammation are closely linked with a greater adherence to healthier diets, such us the Mediterranean diet [90,91,92].

Diet plays an important role in the development or prevention of many chronic diseases [93,94], and may regulate chronic inflammation, improving quality of life [95,96,97]. Thus, dietary composition is able to modulate epigenetic markers such as changes in DNA methylation, the histone or chromatin remodelling of key inflammatory genes, and ncRNAs that may be causal for the development of chronic diseases or beneficial against inflammation; in this way, it can block, retard, or reverse pathologic processes [98,99,100,101,102].

A diet with high a dietary inflammatory index (DII) score has been associated with severe pain and lower quality of life in patients with knee OA [103,104]. Another study showed that the energy-adjusted DII (E-DII) score was associated with a high risk of knee OA in the osteoarthritis initiative (OAI) cohort [105]. The DII has been used to predict inflammatory biomarkers [103,106]. Biomarkers of inflammation, especially serum C-reactive protein (CRP), IL-6, TNF-α and MMPs, have been associated with pain and the progression of OA [107,108,109,110]. Dyer and collaborators showed that biomarkers of inflammation and cartilage degradation related to OA were lower with greater uptake of the Mediterranean diet [111]. In addition, several studies have found that a better quality of life is associated with a higher adherence to this diet [112,113,114,115]. Veronesse and collaborators, in a large cohort of North Americans from the OAI database, demonstrated that a greater adherence to the Mediterranean diet is associated with better quality of life, which is correlated with less pain, disability and depression, better cognitive performance, and better physical functioning [116]. The adherence to the Mediterranean diet was assessed in these studies according to the Mediterranean diet score by established Panagiotakos [117], based on a food frequency questionnaire [118]. Strikingly, greater adherence to the Mediterranean diet is associated with a lower prevalence of knee OA [119]. A high adherence to this diet increases the antioxidant levels in serum samples, with a reduction in oxidative stress biomarkers levels [120,121], such as F2-isoprostane, an indicator of oxidative stress in plasma [122]. Moreover, Martín-Núñez and collaborators found a correlation between lower adherence to the Mediterranean diet pattern and changes in DNA methylation levels and diseases [123].

## 4. Bioactive Compounds: Health-Protective Benefits

The complex biological activities of plants can promote their abundance in secondary metabolites or bioactive compounds, and they are also known as phytonutrients or nutraceuticals. These bioactive compounds are widely known for their unique medicinal properties; they possess antimicrobial [124], anti-inflammatory [125], antiviral [126,127], cardioprotective [128], neuroprotective [129], chemopreventive [130], phytohormone [131], and antioxidant properties [132]. Multiple pathological processes are involved in the pathogenesis of OA, such as inflammation, oxidative stress, apoptosis, autophagy and senescence; hence, phytochemical or bioactive compounds have been used as therapeutic and nutraceutical agents, showing their antiarthritic potential. They mainly exert anti-inflammatory effects through the blockade of pro-inflammatory cytokines (IL1-β, IL-6, IL-8, TNF-α), the inhibition of the NF-κB pathway, antiapoptotic effects, the prevention of oxidative damage to proteins and DNA (reduction in both reactive oxygen species (ROS) and reactive nitrogen species), suppression of the production of prostaglandins and leukotrienes, and reductions in levels of MMPs [133,134,135,136,137].

Bioactive phytochemicals feature a wide variety of compounds, and are classified into phenolics, alkaloids, organosulfur compounds, terpenes and terpenoids, among others, with each class divided into further classes (Figure 1). They are present in fruits, vegetables and spices, and can modify metabolic, cellular, molecular, and epigenetic processes [138]. Polyphenols represent the largest and most ubiquitous group of natural phytochemicals structures; these compounds are present in fruits, vegetables, cereals, tea, dark chocolate, cocoa powder, coffee, extra virgin oil, and wine [139,140,141]. The main groups of polyphenols are flavonoids, phenolic acids, and secoiridoids, among others. Flavonoids a lone comprise more than 10,000 natural compounds, including anthocyanidins, proanthocyanidins flavones, flavanones, flavonols, isoflavones and flavan-3-ols [142,143,144,145].

In this review, a total of 85 bioactive compounds and nutraceuticals with potential anti-OA properties were analysed for use in the management, treatment, or prevention of OA in both humans (Table 1) and animals (Table 2).

In OA, most studied bioactive compounds are curcuminoids [164,165,166,167,168,169,170,171,172,173,174,175,176,177,178,179,180,181,182,183,184,274,275,276,277,278,279], epigallocatechin-3-O-gallate [187,188,189,190,191,192,193], hydroxytyrosol [208,209,210,296], icariin [298,299,300,301,302], oleuropein [223,224], resveratrol [228,229,230,231,232,233,234,321,322,323,324] and sulfuronate [238,239,240,241,242,243]. The most common effects founded in vitro are related to decreased inflammatory and cartilage degradation markers, like MMPs, NO, PGE2 or ROS. On the other hand, in vivo effects observed in OA-induced animal models are critically linked to the reduction in symptoms at the joint level (cartilage, synovium and subchondral bone). Finally, case studies were carried out in humans, showing alleviated pain and enhanced quality of life among other symptoms. Several case studies showed interesting results compared to the conventional analgesic therapy taken by OA patients, especially curcuminoids. It has been proven that they can be as efficacious as ibuprofen [168,169], show potential beneficial effects when used as an adjuvant therapy with diclofenac [170] and meloxican [229] and an alternative therapy for those intolerants to diclofenac’s side effects [171], reduce the use of NSAIDs and gastrointestinal complications [178], and lower adverse effects compared to diacerhein [205,206].

Regarding bioactive compounds’ applications, there are important considerations to take into account: (i) it will be crucial to increase their stability and bioavailability, especially for clinical applications; (ii) a deep understanding must be developed of the underlying molecular mechanisms to increase their bioactivity; and (iii) we must investigate their long-term toxicity and possible side effects.

## 5. Nutritional Epigenomics: Bioactive Compounds in Dietary Balance and Health

Nutritional epigenomics is exceptionally important because it holds great potential in the prevention, suppression and therapy of a wide variety of diseases by altering various epigenetic factors. This novel field involves the lifelong remodelling of our epigenomes, even during cellular differentiation in embryonic and foetal development, by nutritional factors; it also describes how the bioactive molecules can influence and modify gene expression at the transcriptional level [336,337,338,339]. For example, DNA methylation depends on the methyl group donors and cofactors found in foods, thus dietary excess or deficiencies in a critical and sensitive period like embryogenesis can alter the methylation process and gene expression, and therefore the metabolism and physiology of the individual, programming pathologic processes during a lifetime [340,341]. Jirtle and Skinner observed that hypermethylating dietary compounds could reduce the effects of environmental toxicants that cause DNA hypomethylation [342]. An interesting study on *Apis mellifera*, into the different honeybee phenotypes, demonstrated that silencing *Dnmt3* gene expression decreased methylation in the *dynactin p62* gene in larval heads, which led to an increase in the number of queens and a reduction in the number of workers; these epigenetic changes in DNA methylation depended on whether they were fed royal jelly or beebread [343].

Wolff and collaborators provided some of the first evidence that maternal nutrition can impact the epigenome and phenotype of the offspring of dams fed with folate-supplemented diets; this nutrition affected *agouti* gene expression in *A^vy^/a* mice and caused a wide variation in coat colour, ranging from yellow (unmethylated) to light brown (methylated). Pseudoagouti *A^vy^/a* brown mice were lean, healthy, and longer-lived than their yellow phenotype siblings (larger, obese, hyperinsulinemic, more susceptible to cancer) [344]. Furthermore, in macaques that were fed a high-fat diet during pregnancy (predisposing offspring to metabolic syndrome), foetal offspring had increased H3 acetylation and decreased *Hdac1* gene expression in the liver compared to macaques fed with a low-fat diet [345]. An experimental study in Agouti *A^vy^/a* mice fed with genistein (a soy polyphenol), which acts during early embryonic development, showed that genistein-induced hypermethylation persisted into adulthood, by altering the epigenome, decreasing ectopic *agouti* expression, and protecting offspring from obesity, diabetes, and cancer across multiple generations [346]. In addition, experimental data have shown that the maternal consumption of dietary polyphenols such as resveratrol during preconception, gestation and lactation ameliorated metabolic programming. Resveratrol reduced renal oxidative stress, activated nutrient-sensing signals, modulated gut microbiota, and prevented associated high-fructose-intake-induced programmed hypertension in the rat offspring [347].

The four primary targets for epigenetic therapy are DNMTs, HDACs, HATs and miRNA; thereby, numerous bioactive compounds such as sulforaphane, tea polyphenols, ellagic acid, genistein, curcumin, hydroxytyrosol, resveratrol, organosulfur compound, oleanolic acid, and alkaloids have been studied as potent agents for regulating epigenetic mechanisms [102,339,348]. Bioactive compounds can influence epigenetic processes through different mechanisms that interfere with the 1-carbon metabolism and affect S-adenosyl methionine (SAM) levels, meaning they are able to modulate DNA and histone methylation [349]. Many polyphenols, such as quercetin, fisetin, and myricetin, inhibit DNMT by decreasing SAM and increasing S-adenosyl-L-homocysteine (SAH) and homocysteine levels [350].

Global DNA hypomethylation has been associated with the hypermethylation and inactivation of specific genes [351], thus the hypermethylation of cytidine by DNMTs usually results in transcriptional gene silencing and gene inactivation, including of tumour-suppressor genes, while promoters of transcriptionally active genes typically remain hypomethylated [352]. Genes such as *O^6^*-methylguanine methyltransferase, retinoic acid receptor β (*RARB*), the tumour-suppressor *p16^INK4a^*, and the DNA repair gene human *mutL* homologue 1 (*hMLH1*) were shown to be frequently inactivated by hypermethylation, and polyphenols such as epigallocatechin-3-gallate and genistein from soybean were demonstrated to be strong DNMT-inhibitors, leading to the demethylation and reactivation of methylation-silenced genes [353]. DNMTs do not act alone, and they also recruit HDACs to synergistically repress gene transcription [354].

The combination of bioactive compounds acting as DNMT inhibitors, together with phytochemicals that can alter histone modifications, and those that can influence miRNAs expression in OA, are all potentially more synergistic and significant approaches when used as therapeutical strategies to prevent and treat various diseases, including cancer [355,356]. In this context of nutriepigenomics, we have specifically analysed the epigenetic mechanisms related to 12 bioactive compounds, focusing on the prevention or treatment of OA in both humans (Table 3) and animals (Table 4).

Few (but insightful) studies have shown the epigenetic effects of bioactive compounds in OA. The majority of studies are focused on curcuminoids [377,378,379], epigallocatechin-3-O-gallate [361,362,363], hydroxytyrosol [365,366,367,368,381], oleanoic acid [369,370] and resveratrol [372,373,374,375,385,386,387]. By far, the most studied epigenetic mechanisms are miRNAs, which are generally linked to the regulation of inflammatory and cartilage degradation markers. Sirtuins are also well explored in the context of OA.

## 6. Conclusions

In this review, we analysed the importance of bioactive compounds as epigenetic modulators in the prevention and treatment of OA. The reduction in inflammation, as well as catabolic and oxidative activity, is essential in OA treatment. Bioactive compounds or nutraceuticals can directly protect and repair DNA damage, modulating signalling pathways and genes implicated in OA pathogenesis or modifying intra- and extracellular activities. Bioactive compounds are potentially capable of reversing the phenotype of OA chondrocytes. Moreover, the combination of bioactive compounds that act as DNMT inhibitors together with HDAC inhibitors, HAT inhibitors or activators, and miRNA regulators offer more synergistic potential approaches with significance in preventing and treating OA (Figure 2).

Several mixtures have also demonstrated the additive and synergistic potential of bioactive compounds; these mixtures enhanced their chondroprotective properties via anti-inflammatory mechanisms, and reducing oxidative stress. Bioactive compounds are also effective in reducing pain and decreasing the need for NSAIDs, with fewer adverse effects that provide safety and therapeutic efficacy in OA treatment. In addition, new formulations of bioactive compounds have been developed for example with nanoparticles; these phytonutraceuticals possess higher absorption and bioavailability and, could serve as a therapeutic strategy in the prevention and treatment of OA. However, the potential of bioactive compounds as epigenetic regulators in OA has been little studied; further research is needed towards this promising area of research. For this reason, the proposal nutriepigenomic arises and focusses on the ability of numerous bioactive compounds as an alternative to prevent or treat OA.

Future perspectives of bioactive dietary compounds in OA are mainly preventive more than therapeutic. Mostly because the effects of these natural products probably are very small during short periods of time; however, they could be effective when consumed continuously as part of the diet. This indeed could be crucial for a disease like OA, where prevention before symptoms appear is key to stop the progression of the disease. Finally, it will be critical to identify biomarkers to test the efficacy of bioactive compounds at both inter-individual and population levels.

## Figures and Tables

**Figure 1 pharmaceuticals-17-01148-f001:**
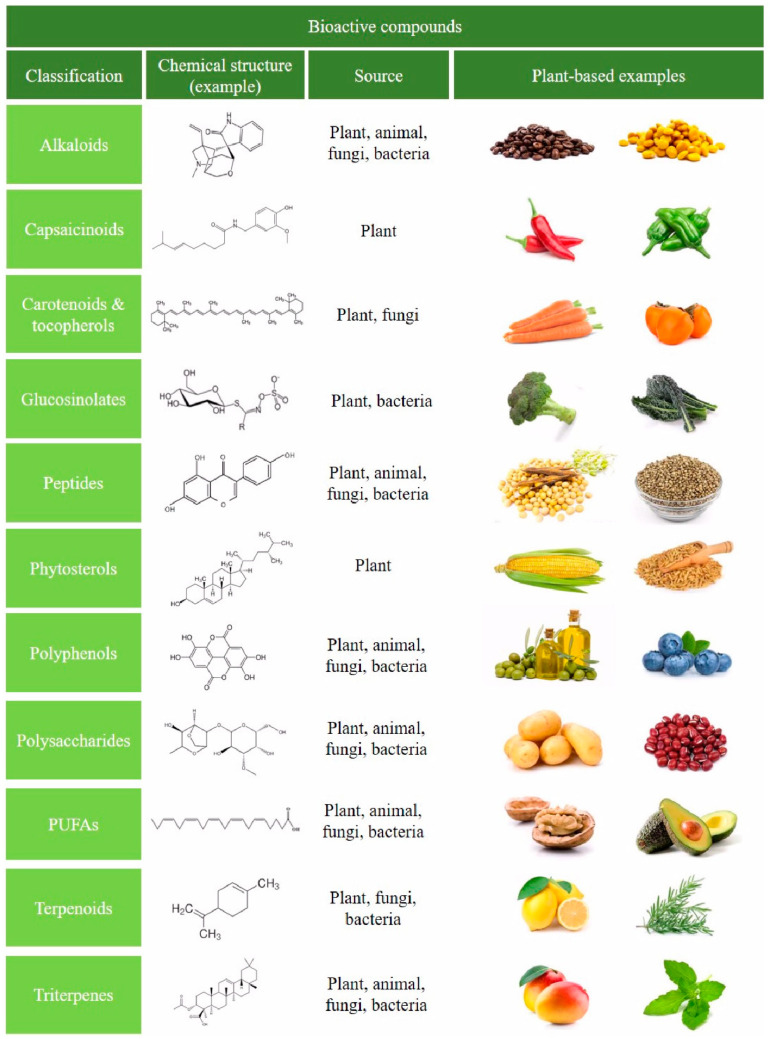
Schematic representation of the classification of the main bioactive compounds in foods. Representative plant-based foods are shown, as well as sources and an illustrative chemical structure example.

**Figure 2 pharmaceuticals-17-01148-f002:**
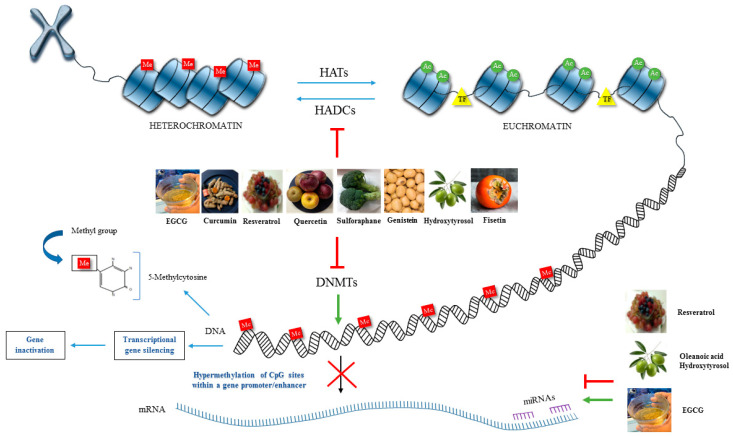
Schematic representation of the impact of bioactive compounds on the main epigenetic mechanisms happening in OA. Several nutraceuticals have been considered as natural epigenetic modulators that can modify the activity of various epigenetic factors (DNA methylation, HATs, HDACs and miRNA) and, altering the expression of genes related to inflammation and cartilage destruction, being potentially able to reverse the phenotype of OA chondrocytes.

**Table 1 pharmaceuticals-17-01148-t001:** Bioactive compounds and nutraceuticals used for the management, treatment, or prevention of OA in humans.

Bioactive Compounds	Sources/Classes	Effects of Bioactive Compounds	Ref.
ALM16 Herbal mixture Major active compounds: (calycosin, calycosin-7-*O*-β-d-glucopyranoside) lithospermic acid	Dried roots of: (*Astragalus membranaceus*) Isoflavonoids (*Lithospermum* *erythrorhizon*) Phenolic acid	Effects in IL-1β-stimulated SW1353 chondrocytes: Prevented glycosaminoglycan degradation Decreased MMP-1, MMP-3 and MMP-13 levels	[146]
Anthocyanidins: (Cyanidin-3-glucoside, pelargoni din3-glucoside) Flavonols: (Quercetin, kaempferol, mirycetin) Flavanols: (Epigallocatechin 3-gallate, catechin) Ellagitannins	(*Fragaria ananassa*) Strawberry (*Vaccinium corymbosum*) Blueberry (*Punica granatum* L.) pomegranate Approx. 40 phenolic compounds identified: Flavonoids Tannins	Effects in obese patients with knee OA: Alleviated pain and enhanced quality of life Decreased inflammatory and cartilage degradation markers Decreased IL-6, IL-1β, and MMP-3 levels in blood samples	[147]
		Effects in knee OA patients: Decreased pain and stiffness and improved gait performance and quality of life Improvement in daily physical activities	[148]
		Effects in OA chondrocytes: Suppressed the IL-1β-induced activation of RUNX-2, MKK3/6 and p38-MAPK isoforms in chondrocytes derived from OA cartilage	[149]
		Effects in IL-1β-induced OA chondrocytes: Downregulated *MMP1, MMP3,* and *MMP13* mRNA expression Inhibited activation of APKs and the DNA-binding activity of NF-κB	[150]
Arctigenin (Phenylpropanoid dibenzylbutyrolactone)	*Arctium lappa* Greater burdock Lignan	Effects in IL-1β-induced OA chondrocytes: Decreased ECM degradation Enhanced ECM synthesis and upregulated COL2A1 and ACAN Downregulated MMP-13 and ADAMTS-5 Decreased *IL6, NOS2*, *TNFA* and *COX2* in mRNA and protein expression Inhibition of NF-κB/PI3K/Akt signalling pathway	[151]
Astragalin (kaempferol 3-glu-coside)	Leaf extract of: *Rosa agrestis* Flavonoids	Effects in IL-1β-induced chondrocytes: Inhibited inflammatory responses Inhibited NO, PGE2, NF-κB, ERK1/2, JNK, and p38 MAPK production by PPAR-γ activation in a dose-dependent manner	[152]
Avocado/Soybean Unsaponificables ASU (β-sitosterol, campesterol, and stigmasterol) Triterpenes	*Persea gratissima* and *Glycine max* Mixture of avocado and soybean unsaponifiables (Phytosterols) Triterpene alcohols	Effects in IL-1β-induced OA chondrocytes: Promoted cartilage repair Inhibited IL-6, IL-8, MIP-1β, MMP-3, NO, and PGE2 production Stimulated TIMP-1, TGF-β1, and ACAN production	[153]
		Effects in OA subchondral osteoblasts/OA chondrocytes: Promoted regulation of anabolic and catabolic processes Downregulated ALP, OC, and TGF-β1 levels Prevented inhibition of ECM components (*COL2A1* and *ACAN* mRNA expression)	[154]
		Effects in LPS-stimulated monocyte/macrophage-like cell associated with the synovial membrane: Showed anti-inflammatory effects Supressed *TNFA*, *IL1B, COX2, NOS2* gene expression Downregulated PGE2 and nitrite production	[155]
		Effects in chondrocytes: Attenuated inflammatory response at both gene transcription and protein levels Reduced G-CSF, RANTES and PGE2 levels induced by LPS Increased 12,13-DiHOME	[156]
Baicalin	(*Scutellaria baicalensis* *Georgi*) Mainly extracted from dry root Flavone glycoside (flavonoid)	Effects in IL-1β-induced OA chondrocytes: Reduced *COX2, NOS2, MMP3, MMP13* and *ADAMTS5* gene expression via inhibition of NF-κB activation Inhibited NO and PGE2 production Inhibited the downregulation of *ACAN* and *COL2A1* mRNA	[157]
Berberine	Medicinal herbs: *Hydrastis canadensis* *Berberis aristate* *Cortex phellodendri* *Coptis chinensis* Isoquinoline-derivative alkaloid	Effects in OA synovial fibroblast: Attenuated CCN2-induced IL-1β expression, via inhibition of ROS-related ASK1, p38/JNK, NF-κB signalling pathways	[158]
Butein	*Rhus verniciflua* stem bark of cashews and the genera Dahlia, Butea, Searsia (Rhus) and Coreopsis are common sources Chalcones (flavonoids)	Effects in IL-1β-induced OA chondrocytes: Reduced IκB-α degradation and NF-κB p65 activation Downregulated *COX2, NOS2, IL6, TNFA, MMP13* gene and protein expression Inhibited *MMP1, MMP3, ADAMTS4* and *ADAMTS5* mRNA expression Reduced the degradation of *COL2A1* and *SOX9* mRNA and protein expression Downregulated NO and PGE2 production	[159]
Casticin (Vitexicarpin)	*Vitex rotundifolia* L. Polymethoxyflavonoid	Effects in IL-1β-induced OA chondrocytes: Prevented inflammation by inhibition of NF-κB signalling pathway Decreased NO, PGE2, TNF-α, IL-6, MMP-3, MMP-13, ADAMTS-4 and ADAMTS-5 production Inhibited *NOS2* and *COX2* mRNA and protein expression Increased *ACAN* and *COL2A1* mRNA expression	[160]
Celastrol	(*Tripterygium wilfordii* *Hook F.*) root bark “Thunder of God Vine” Pentaciclic Triterpenes	Effects in IL-1β-induced OA chondrocytes: Suppressed the activation of NF-κB in human osteoarthritic chondrocytes Inhibited *HSP90B, COX2, NOS2, MMP1, MMP3, MMP13* mRNA and protein expression Decreased NO and PGE2 levels	[161]
Cinnamophilin	(*Cinnamomum philippinense*) Extracted from the root Lignan	Effects in IL-1β-stimulated SW1353 chondrocytic cell line: Showed chondroprotective properties against collagen matrix breakdown Inhibited MMP-1 and MMP-13 activity via inhibition of NF-κB, JNK, ERK, and p38 MAPK Inhibited IκB-α degradation, and IKK-α/β and p65 phosphorylation Blocked the activity of c-Jun by inhibition of JNK	[162]
Cryptotanshinone	(*Salvia miltiorrhiza* *Bunge*) Extracted from the root of the plant Diterpene quinones	Effects in IL-1β-induced OA chondrocytes: Inhibited inflammation by suppression of nuclear translocation of NF-κB p65 and MAPK activation Inhibited phosphorylation of IκB, IKKα/β and IκBα degradation Suppressed NO, PGE2, IL-6, TNF-α, NOS2, COX-2, MMP-3, MMP-13, and ADAMTS-5 levels	[163]
Curcuminoids: Curcumin Demethoxycurcumin, Bisdemethoxycurcumin	(*Curcuma longa*) (*Curcuma domestica*) Turmeric rhizome Diarylheptanoids (Phenolic compounds)	Effects in IL-1β-induced chondrocytes: Protected against catabolic effects Inhibited suppression of COL2A1 synthesis Inhibited NF-κB signalling pathway and prevented its translocation to the nucleus Inhibited MMP-3 synthesis	[164]
		Effects in IL-1β-induced chondrocytes: Demonstrated chondroprotective, antiapoptotic and anti-catabolic properties Inhibited cell degradation Inhibited suppression of COL2A1 Increased β1-integrin receptors synthesis Decreased caspase-3 activation (antiapoptotic effect)	[165]
		Effects in chondrocytes: Demonstrated anti-inflammatory effects stimulated by IL-1 and TNF-α Suppressed NF-κB activation and inhibited p65 phosphorylation and nuclear translocation Blocked the IκBα phosphorylation and degradation Inhibited IL-1β-induced Akt phosphorylation Inhibited COX-2 and MMP-9 synthesis	[166]
		Effects in IL-1β-induced OA chondrocytes/OA cartilage explants: Demonstrated anti-inflammatory activity Suppressed ECM degradation Inhibited MMP-3, PGE2, NO, IL-6, and IL-8 production	[167]
		Effects in knee OA patients: Showed that *C. domestica* extracts were as efficacious as ibuprofen Demonstrated pain reduction and functional improvement Showed fewer gastrointestinal adverse effects than ibuprofen	[168]
		Effects in knee OA patients: Enhanced knee functions and reduced knee pain Demonstrated the efficacy and safety of curcumin extract 2000 mg/day was equivalent to ibuprofen 800 mg/day for 6 weeks therapy	[169]
		Effects in knee OA patients: Showed potential beneficial effects as an adjuvant therapy with diclofenac in knee OA Showed additive improvements in decreasing pain Reduced inflammation without increasing the side effects in comparison with diclofenac alone	[170]
		Effects in knee OA patients: Proved to be a substitute treatment option in knee OA patients who are intolerant to the side effects of diclofenac Demonstrated gastroprotective and antiulcer effects, compared with the adverse effects of non-steroidal anti-inflammatory drugs	[171]
		Effects in IL-1β-induced temporomandibular joint chondrocytes: Showed anti-inflammatory, antioxidant, and cartilage-protective effects by activating the NRF2/ARE (HO-1, SOD2, NQO-1, and GCLC) pathway Inhibited *NOS2*, *COX2*, *IL6*, *MMP1*, *MMP3*, *MMP9*, *MMP13*, *ADAMTS4* and *ADAMTS5* mRNA and protein levels Increased *COL2A1* and *ACAN* mRNA expression	[172]
Curcumin nanoparticles	Topical treatment	Effects in IL1β-induced chondrocytes: Enhanced chondroprotective properties against the production of inflammatory and catabolic mediators Reduced *IL1B, TNFA, ADAMTS5, MMP1,* *MMP3*, and *MMP13* mRNA expression Increased levels of the chondroprotective transcriptional regulator *CITED2* gene	[173]
Combination: Curcumin with resveratrol	Resveratrol (*trans*-3, 4′- trihydroxystilbene)	Effects in IL-1β-induced chondrocytes: Inhibited inflammatory and catabolic effects and activated β1-integrin and Erk1/2 Demonstrated synergistic effects in suppressing apoptosis	[174]
Theracurmin	Highly bioavailable form of curcumin (A surface-controlled water-dispersible form of curcumin)	Effects in knee OA patients: Showed high bioavailability that was 27-fold higher than that of curcumin powder without adverse effects	[175]
		Effects in knee OA patients: Showed high absorption and enhanced chondroprotective effects Reduced pain and decreased NSAID necessity Demonstrated anti-inflammatory effects Showed therapeutic efficacy and safety (180 mg/day orally for six months)	[176]
RA-11 (Nutraceutical mixture)	*Curcuma longa* (*Withania somnifera*) Ashwagandha Terpenoids, flavonoids, tannins, alkaloids (*Boswellia serrata*) Olibano Boswellic acids (terpenoid) (*Zingiber officinale*), Ginger Phenolic and terpene compounds	Effects in knee OA patients: Demonstrated greater potency, efficacy, and excellent security for OA treatment over 32 weeks of therapy Showed significant reduction in the pain VAS and the modified WOMAC index scores (pain, stiffness, and physical function difficulty)	[177]
Phytosome complex (Meriva)	Curcuminoid mixture with phosphatidylcholine (soy lecithin, a phospholipid)	Effects in OA patients: Improved oral absorption and bioavailability Reduced all WOMAC scores after eight months of treatment with 200 mg curcumin/d Decreased inflammatory markers sCD40L, IL-1β, IL-6, sVCAM-1, and ESR Decreased use of NSAIDs/painkillers and gastrointestinal complications Improved emotional functions and quality of life	[178]
		Effects in IL-1β-induced HCH-c chondrocytes: Improved the solubility of curcumin and enhanced the chondroprotective effect via the anti-inflammatory system in chondrocytes Suppressed *MMP1, MMP2, MMP3, MMP9, MMP13, NOS2* and *COX2* mRNA expressions Inhibited TNF-α, IL-1β, IL-6, IL-8 and PGE2 levels	[179]
Mixture: Curcuminoids Hydrolysed collagen and Epigallocatechin-3-gallate	(*Curcuma longa* L.) Turmeric rhizome Polyphenols Hydrolysed collagen (High levels of glycine and proline, amino acids for the stability and regeneration of cartilage) (*Camellia sinensis*) Green tea Epigallocatechin-3-gallate (flavanol)	Effects in IL-1β-induced OA chondrocytes: Showed additive and synergistic effects Demonstrated to be significantly more efficient in inhibiting inflammation and catabolic processes Suppressed NF-κB activation and its translocation to the nucleus via inhibition of phosphorylation and degradation of IκBα and p65 phosphorylation Inhibited MMP-3, IL-6, NO production	[180]
Combination: Curcumin Flavocoxid: baicalin and catechin β-caryophyllene	(*Curcuma longa*) Phenolic compounds (*Scutellaria baicalensis,* Baikal skullcap) and (*Acacia catechu,* catechu) Baicalin and catechin Flavonoids (*Copaifera spp,* copaiba) and (*Cannabis* spp., marijuana/hemp) β-caryophyllene, a (bicyclic sesquiterpene)	Effects in LPS and IL-1β-stimulated chondrocytes: Demonstrated anti-inflammatory activity and safety and did not affect cell viability in chondrocytes Reduced *IL1B* mRNA in a dose-dependent manner Showed strong synergy potential for OA treatment Reduced the mRNA expression of transcription factors *NFKB* and *STAT3* Increased *COL2A1* mRNA expression	[181]
Botanical formulation (Mixodin): Curcumin, Gingerols, and Pyrene	(*Curcuma longa*) Turmeric Phenolic compounds (*Zingiber officinale*) Ginger Gingerols Phenolic compounds (*Piper nigrum*) Black pepper Pyrene (Alkaloid)	Effects in knee OA patients: Showed synergic, anti-inflammatory and hypoalgesic effects in chronic knee OA (twice a day for 4 weeks) Observed as a safe alternative to chemical drugs, with lower adverse effects than Naproxen Decreased PGE2 levels in blood samples (curcumin 300 mg, gingerols 7.5 mg, and piperine 3.75 mg) similar to Naproxen drug (250 mg twice/day)	[182]
Botanical composition NXT15906F6: ethanol/aqueous extract of tamarind seed (proantocyanidins) and aqueous ethanol extract of turmeric (curcuminoids) NXT19185: (combination of NXT15906F6 plus an aqueous ethanol extract of mangosteen (α-mangostin, β-mangostin, and γ-mangostin) and (epicatechin and quercetin)	*Tamarindus indica* Tamarind seeds Polyphenols *Curcuma longa* *Garcinia mangostana* fruit rind Polyphenolic xanthones Flavonoids	Effects in knee OA patients/serum/urine: NXT15906F6 (250 mg) or NXT19185 (300 mg) daily for 50–6 days Decreased inflammatory processes, joint pain and stiffness Improved musculoskeletal function Inhibited TNF-*α*, IL-6, MMP-3 and CRP levels in serum Protected against cartilage erosion Reduced CTX-II (a cartilage degradation marker) in a urine sample Reduced WOMAC, VAS, stair climb test scores Improved Lequesne’s functional index, the 6-min walk test and knee flexion range of motion scores	[183]
Botanical composition (LI73014F2 2:1:2 ratio): Gallic acid, chebulagic acid, chebulic acid, chebulinic acid, gallotannins, ellagitannins (punicalagin), ellagic acid Diferuloylmethane Demethoxycurcumin Bisdemethoxycurcumin, and turmeric acid Boswellic acids: 3-O-acetyl-11-keto-β-boswellic acid, 11-keto-β-boswellic acid, and β-boswellic acid	(*Terminalia chebula*) fruit myrobalan Tannins (polyphenols) (*Curcuma longa*) Polyphenols (*Boswellia serrata*) Olibanum Pentacyclic triterpenes	Effects in IL-1β-induced HCHs chondrocytes: Reduced inflammation and apoptosis via NF-κB/MAPK signalling pathway inhibition Inhibited pro-inflammatory mediators (COX-2, 5-LOX, and metabolic pathways products mPGES-1, PGE2, and LTB-4) Decreased IL-1β, TNF-α, IL-6, MMP-2, MMP-3, MMP-9 and MMP-13 protein levels Provided therapeutic efficacy in OA management by reducing cartilage damage	[184]
Delphinidin	Pomegranate, berries, dark grapes, aubergine, tomato, carrot, purple sweet potatoes, red cabbage, and red onion Anthocyanidin (Flavonoid) Delphinidin, the most abundant anthocyanidin present in pomegranate fruit extract (*Punica granatum*)	Effects in IL-1β-induced OA chondrocytes: Inhibited phosphorylation of IκB, IKKα/β, NIK, IRAK1 Inhibited *COX2* mRNA and protein expression and PGE2 production via suppression of NF-κB activation Downregulated *IKKB* mRNA and protein expression	[185]
Ellagic acid	Fruit peel of raspberries, strawberries, cranberries, pomegranate, walnuts, pecans, grapes Dimeric derivative of gallic acid Phenolic compound	Effects in IL-1β-induced OA chondrocytes: Inhibited inflammation and ECM loss Upregulated COL2A1 and ACAN Suppressed NF-κB p65 activation Decreased NO, PGE2, IL-6, TNF-α, ADAMTS-5 and MMP-13 in a dose-dependent manner Inhibited *NOS2, COX2* mRNA and protein expression	[186]
Epigallocatechin-3-O-gallate	*Camellia sinensis* Green tea Flavan-3-ols or flavanols (Flavonoids)	Effects in IL-1β-induced chondrocytes: Showed anti-inflammatory and anti-catabolic effects in a dose-dependent manner Inhibited *MMP1* and *MMP13* mRNA and protein expression Inhibited NF-κB and AP1 levels Effects in cartilage explants: Inhibited cartilage matrix degradation Downregulated glycosaminoglycans release	[187]
		Effects in IL-1β-induced OA synovial fibroblasts: Showed efficacy in the control of inflammation Inhibited *COX2* mRNA and protein expression Supressed PGE2 and IL-8 production	[188]
		Effects in IL-1β-induced OA chondrocytes: Decreased *NOS2* mRNA and protein expression and NO production Inhibited NF-κB p65 activation and translocation to the nucleus by suppressing the degradation of its inhibitory protein IκBα in the cytoplasm	[189]
		Effects in IL-1β-induced chondrocytes: Antioxidant properties against cytotoxicity Inhibited ROS release and accumulation from both intracellular and extracellular environments Inhibited PGE-2, NO, COX-2 and NOS2 production	[190]
		Effects in IL-1β-induced OA chondrocytes: Inhibited catabolic mediators of cartilage degradation Inhibited JNK isoforms phosphorylation and activation Blocked c-Jun phosphorylation in the cytoplasm and reduced the DNA binding activity of AP-1 in the nuclei	[191]
		Effects in OA chondrocytes: Suppressed the AGE-induced *TNFA* and *MMP13* mRNA and protein expression Inhibited AGE-BSA-induced degradation of IκBα and nuclear translocation of NF-κB p65 Inhibited MAPK and NF-κB activation	[192]
		Effects in IL-1β-stimulated OA chondrocytes: Showed anti-inflammatory activity Inhibited NF-κB and MAPKs pathway Inhibited *TRAF6* mRNA and protein expression Downregulated *IL6, IL8, TNFA, IL1B, IL7* and *GMCSF* mRNA and protein expression Blocked *ENA78, GRO, GROA, MCP1, MIP1B, MIP3A, GCP2, IP10* and *NAP2* chemokines’ expression	[193]
Fatty acids n-3 PUFAs omega 3 polyunsaturated fatty acids	Soybean, canola, olive oils, flaxseed, walnuts, marine phytoplankton and fish oil ALA: α-linolenic acid EPA: eicosapentaenoic DHA: docosahexaenoic	EPA decreased *MMP3* and *MMP13* mRNA EPA decreased chondrocyte apoptosis by inhibiting oxidative stress-induced phosphorylation of p38 MAPK and p53	[194]
Genistein	(*Gycine max*) soybean Isoflavone (flavonoids)	Effects in LPS-induced chondrocytes: Suppressed COX-2 and NO protein levels in a dose-dependent manner Reduced IL-1β and YKL-40 (a marker of cartilage degradation) levels	[195]
		Effects in IL-1β-induced OA chondrocytes: Reduced inflammation and oxidative stress Decreased MMP-1, MMP-3, MMP-13, MMP-9, NO, COX-2, NOS2 Stimulated HO-1 associated with NRF-2 pathway activation	[196]
		Effects in IL-1β-induced chondrocytes: Upregulated COL2A1, ACAN and ERα protein expression in a dose-dependent manner Inhibited apoptosis Reduced caspase-3 and TNF-α levels	[197]
Gingerols Shogaols	*Zingiber officinale* and *Alpinia galanga* Phenolic compounds	Effects in knee OA patients: Demonstrated improvements in WOMAC index and VAS pain profiles (6 weeks treatment 225 mg/twice day) Showed a good safety profile with mostly mild gastrointestinal side effects	[198]
		Effects in knee OA patients: Reduced inflammatory markers (1 g/d for 3 months) Decreased CRP and NO in serum and improved pain and mobility	[199]
Gingerols and shogaols + isobutylamides and 2- methylbutylamimide	Highly standardized ginger and echinacea extract *Zingiber officinale* *Echinacea angustifolia* Roots (alkylamides: fatty acid amides)	Effects in knee OA patients: Showed anti-inflammatory, synergistic properties during four-week supplementation Reduced chronic pain and improved knee function Showed to be secure without important side effects Could be an alternative for NSAIDs non-responders	[200]
Gingerols Shogaols Nanoparticles	*Zingiber officinale* ginger extract in nanostructure lipid carrier	Effects in knee OA patients: Decreased stiffness and the reduction in pain was significantly greater than compared to topical diclofenac (12 weeks treatment) Improved physical function	[201]
Gingerols, shogaols and Spilanthol (MITIDOL)	*Zingiber officinale* *Acmella oleracea* Sphilantol (alkamide) food-grade lecithin formulation of standardized extracts	Effects in knee OA patients: Inflammatory markers’ reduction (CRP and erythrocyte sedimentation rate) Antioxidant and analgesic properties Improved knee function and free of side effects	[202]
Harpargoside, Harpagide y Procumbide β-cariofileno, α-humuleno y α-copaeno Oleanolic acid, Ursolic acid and 3β-acetyloleanolic acid Eugenol Acteoside and Isoacteoside	*Harpagophytum* *procumbens* (HP) devil’s claw root HP extract Iridoid glucosides Sesquiterpenes Triterpenes Monoterpene Phenolic glycosides	Effects in fibroblast-like synoviocytes/synovial membrane/OA patients: Showed anti-inflammatory and antinociceptive effects HPE_H2O,_ HPE_DMSO_ increased *CB2* mRNA expression and inhibited PI-PLC β2 isoform expression All the HPE extracts inhibited *FAAH* mRNA expression and enzymatic activity (HPE_EtOH100_ was the most effective)	[203]
		Effects in IL-1β-induced chondrocytes: Suppressed inflammatory cytokines/chemokines Inhibited *IL6,* and *MMP13* mRNA expression Suppressed c-FOS/AP-1 transcription factor	[204]
		Effects in knee and hip OA patients: Showed efficacy and superior safety as a therapeutic agent (2610 mg of powdered cryoground) compared to diacerhein (100 mg/day) for 4 months Showed lower adverse effects than diacerhein	[205,206]
		Effects in IL-1β-induced chondrocytes: Suppressed MMP-1, MMP-3 and MMP-9 production via inhibition of inflammatory cytokines TNF-α and IL-1β synthesis	[207]
Hydroxytyrosol (HT)	*Olea europea* L. Olive leaf extract Fruits Extra virgin oil HT is more abundant in the processed fruit and olive oil Secoiridoid derivative	Effects in knee OA patients: Demonstrated pain inhibition over a 4-week period Decreased pain measurement index (Japanese Orthopedic Association score) and VAS scores HT was considered effective when it reached the knee joint in an unmetabolized form Showed antioxidant and anti-inflammatory properties	[208]
HT and Verbascoside	Verbascoside: Hydroxycinnamic acid derivative (phenolic compound)	Effects in OA chondrocytes: Showed chondroprotective effects and reduced intracellular ROS generation Suppressed oxidative stress via p38 and JNK signalling pathways HT downregulated ICE/caspase-1 indicating a potential anti-inflammatory effect	[209]
Hydroxytyrosol/ Procyanidins (Oleogrape^®^SEED)	Extract from olive and grape seed: (*Olea europea* L.) mainly found in olive leaf and oil Phenolic compound (*Vitis vinifera, grape*) Flavonoids Other sources: pine bark, cocoa, raspberry, vegetables, legumes, nuts	Effects in IL-1β-induced chondrocytes: Demonstrated chondroprotective properties Decreased NO, PGE2, and MMP-13 production Reduced NF-κB p65 signalling pathway Effects of serum enriched with HT/procyanidins metabolites on primary articular chondrocytes stimulated with IL-1β (*ex vivo* methodology): Reduced NO, PGE2, and MMP-13 levels	[210]
Icariin	*Epimedium sagittatum* flavonol glycoside	Effects in OA fibroblast-like synoviocytes: Inhibited inflammatory response, apoptosis, ER stress and ECM degradation Decreased *IL1β, MMP14,* and *GRP78* gene and protein expression	[211]
		Effects in IL-1β-induced SW1353 chondrosarcoma cells: Showed chondroprotective properties and inhibited *MMP1, MMP3* and *MMP13* gene and protein expression via MAPK pathways Inhibited p38, ERK and JNK phosphorylation	[212]
		Effects in IL-1β-induced chondrocytes: Demonstrated chondroprotective and antioxidant functions without cytotoxic effects by activation of *NRF2* mRNA Inhibited ECM degradation and ROS production Promoted *SOD1, SOD2* mRNA and GPX activity Decreased *MMP3, MMP9, MMP13* and *ADAMTS4* mRNA expression	[213]
Indole tetracyclic alkaloids Oxindole alkaloids Indole pentacyclic alkaloid Glycoindole alkaloids Quinovic acids Tannins	*Uncaria guianensis* *Uncaria tomentosa* Cat’s claw alkaloids Triterpenes heterosides polyphenols	Effects in knee OA patients: Showed antioxidants and anti-inflammatory properties Alleviated knee pain and promoted benefit to the joints, tolerability and safety at high concentrations Reduced the toxic side effects of NSAIDs and had no deleterious effects on blood or liver function or other significant side-effect Improved OA management and treatment	[214]
Isofraxidin	*Siberian ginseng* and *Apium graveolens* Coumarin (phenolic compound)	Effects in LPS-induced OA chondrocytes: Decreased iNOS, COX-2, NO, PGE2, TNF-α and IL-6 levels Suppressed ECM degradation Inhibited TLR4/MD-2 complex formation and NF-κB signalling pathway	[215]
		Effects in IL-1β-induced OA chondrocytes: Suppressed inflammatory mediators and ECM degradation through inhibiting the NF-κB pathway Inhibited IκB-α degradation Blocked NO and PGE2 production Inhibited *COX2*, *NOS2*, *MMP1*, *MMP3*, *MMP13*, *ADAMTS4* and *ADAMTS5* mRNA expression and protein levels Increased ACAN and COL2A1 levels	[216]
Juglanin	*Polygonum aviculare* *Juglans regia* L. Diarylheptanoid derivative Flavonoids	Effects in IL-1β-induced OA chondrocytes: Inhibited inflammatory responses through suppressing phosphorylation of NF-κB p65 Suppressed IκBα degradation Inhibited NO, PGE2, IL-6, TNF-α, MMP-1, MMP-3, and MMP-13 levels Decreased *NOS2, COX2, ADAMTS4* and *ADAMTS5* mRNA and protein expression	[217]
Licochalcone A	*Glycyrrhiza glabra*, liquorice root *Glycyrrhiza inflate* Flavonoids	Effects in IL-1β or TNF-α-induced OA chondrocytes: Showed anti-inflammatory properties Inhibited PGE2 and NO production Inhibited MMP-1, MMP-3, and MMP-13 levels Inhibited *NOS2* and *COX2* mRNA expression Inhibited NF-κB activation and IκBα degradation Increased *NRF2* and *HO1* mRNA and protein expression	[218]
Acetylated ligstroside aglycone: (Chemically acetylated version of ligstroside aglycone)	(*Olea europea* L.) Extra virgin olive oil Ligstroside aglycone (p-HPEA-Elenolic acid) Secoiridoids	Effects in IL-1β/OSM-induced OA chondrocytes/OA cartilage: Reduced *NOS2*, *MMP13* gene and protein expression Enhanced anti-inflammatory activity compared to the natural compound ligstroside Inhibited NO levels, proteoglycan (PG) loss and cartilage degradation	[219]
Myrcene	*Eryngium duriaei* monoterpene	Effects in IL-1β-induced chondrocytes: Showed anti-inflammatory and anti-catabolic properties in human chondrocytes Inhibited *NOS2* mRNA expression and activity, and the NF-κB pathway Reduced *MMP1* and *MMP13* gene expression Decreased the phosphorylation of JNK, p38, and ERK1/2 Increased *TIMP1* and *TIMP3* mRNA Decreased *COL1* mRNA and promoted the maintenance of the differentiated chondrocyte phenotype	[220]
Myricetin	*Labisia pumila* *Trigonella foenum graecum* L. *Anacardium* and *Mangifera* species (*Anacardiaceae*) Grapes, berries, chard spinach, broadbeans, garlic, peppers Flavonol	Effects in IL-1β stimulated chondrocytes: Inhibited inflammatory mediators and cytokines and exerted no significant dose-dependent cytotoxicity Inhibited *NOS2* and *COX2* mRNA and protein Decreased NO and PGE2 production Suppressed TNF-α and IL-6 levels Inhibited ECM degradation and inhibited *ADAMTS5* and *MMP13* gene expression Promoted *ACAN* and *COL2A1* gene Inhibited NF-κB p65 nuclear translocation and activation and inhibited IκBα degradation Increased NRF2 translocation into the nucleus and activation, and HO-1 expression in cytoplasm against inflammation response via PI3K/Akt	[221]
Oleocanthal (decarboxymethyl ligstroside aglycone)	(*Olea europea* L.) Fruits, leaves, extra virgin oil Secoiridoid derivative (Phenolic compounds)	Effects in LPS-activated OA chondrocytes: Suppressed inflammation and OA progression Blocked MAPKs/NF-κB pathways Inhibition of NOS2 and NO protein synthesis Inhibited *IL6, IL8, COX2, NOS2, MIP1α, TNFA, LCN2, MMP13* and *ADAMTS5* mRNA expression	[222]
Oleuropein	(*Olea europea* L.) Olive leaves and seeds, pulp and peel of unripe olives, extra virgin oil High amounts in unprocessed olive fruit Secoiridoid (phenolic compounds)	Effects in IL-1β-stimulated OA chondrocytes: Suppressed phosphorylation of NF-κB p65 and nuclear translocation, IκB-α degradation, and MAPK activation Inhibited *COX2, NOS2, MMP1, MMP13*, and *ADAMTS5* mRNA expression Inhibited degradation of ACAN and COL2A1 Inhibited NO and PGE2 production	[223]
		Effects in primary OA chondrocytes (OACs)/human mesenchymal stem cells/synoviocytes/bone cells: Reduced connexin 43 protein expression, gap junction intercellular communication and *TWIST1* mRNA and increased *COL2A1* and *ACAN* mRNA in OACs Reduced inflammatory and catabolic factors *IL1B, IL6, COX2* and *MMP3* mRNA expression and protein levels in OACs Restored chondrocyte phenotype Enhanced osteogenesis and chondrogenesis in hMSCs Improved cartilage and joint regeneration Caused a significant reduction in senescent cells in OACs, synoviocytes and bone cells	[224]
Oleuropein Hydroxytyrosol, Verbascoside, Luteolin, (ZeyEX)	(*Olea europaea* L., olive leaves) Olive leaf extract Polyphenolic compounds	Effects in OA chondrocytes: Inhibited IL-6, IL-1β, and TNF-α and improved COL2A1 levels Inhibited p-JNK/JNK ratio but no effect of ibuprofen Inhibited Casp-1/ICE, ROS, lipid hydroperoxide, 4-Hydroxynonenal-protein adduct, advanced glycation (glycoxidation) end product protein adduct AGE, 3-Nitrotyrosine 3-NT, GM-CSF, COMP, receptor for advanced glycation end product RAGE and TLR4 levels	[225]
Puerarin	(*Radix puerariae*) Root of Pueraria Phytoestrogen (Isoflavone)	Effects in IL-1β-induced OA chondrocytes: Showed antioxidative and anti-inflammatory effects and increased cell proliferation Decreased PGE-2, IL-6 and TNF-α levels Effects in IL-1β-treated monocytes/macrophage: Reduced IL-6, IL-12 and TNF-α expression Increased TGF-β1 and IL-10 levels	[226]
Quercetin	(*Achyranthes bidentata*) Flavonol (flavonoid)	The docking of PIM1-quercetin, CYP1B1-quercetin, and HSPA2-quercetin by *Achyranthes bidentate* were the key targeted proteins of quercetin in the treatment of OA	[227]
Resveratrol	Root extracts of the weed *Polylygonum cuspidatum* *Vitis vinifera* red grapes, blueberries cranberries, peanuts, Stilbenes (polyphenols)	Effects in IL-1β-induced SW1353 cell line: TLR4 inhibition related to PI3K/Akt activation PI3K/Akt activation was attenuated after the TLR-4 pathway was blocked by the TLR-4 inhibitor CLI-095 Unable to reduce TLR4 protein expression after the PI3K inhibitor LY294002 blocked PI3K/Akt signalling	[228]
		Effects in knee OA patients: Demonstrated efficacy and safety as an adjuvant with meloxican during a 90-day period Decreased knee joint pain (dose 500 mg/day) without adverse effects Effects in serum: Decreased biomarkers of inflammation IL-1β, IL-6, TNF-α, CRP	[229]
		Effects in IL-1β-stimulated chondrocytes: Showed chondroprotective effects Suppressed the activation of IL-1β-induced catabolism and apoptosis in human chondrocytes *in vitro* Blocked the downregulation of cartilage matrix marker COL2A1 and the cell matrix receptor β1-integrin protein expression Inhibited caspase-3 activation and PARP cleavage in a time-dependent manner	[230]
		Effects in IL-1β-stimulated chondrocytes: Protected against catabolic effects Inhibited membrane-bound IL-1β and mature IL-1β protein production Inhibited p53 accumulation in a dose-dependent manner and produced degradation of p53 by the ubiquitin-independent pathway Inhibited p53-dependent apoptosis Suppressed ROS, caspase 3 activation, and PARP cleavage	[231]
		Effects in IL-1β-stimulated OA chondrocytes: Blocked mitochondrial membrane depolarization, maintained mitochondrial function and restored ATP levels Inhibited apoptosis via the inhibition of PGE2 through the suppression of *COX2* mRNA and protein expression Reduced (apoptotic markers) cytochrome c release from mitochondria and annexin V Inhibited DNA fragmentation Effects of IL-1β-stimulated OA cartilage explants: Increased PG synthesis Decreased MMP-1, MMP-3, MMP-13 Inhibited PGE2 and leukotriene B_4_ levels	[232]
		Effects in IL-1β-induced SW1353 cells: Demonstrated anti-inflammatory and anti-osteoarthritic properties Inhibited TLR4/NF-кB and inflammatory responses via the inhibition of MyD88-dependent and -independent signalling pathways Decreased IL-6 levels Activated PI3K/Akt pathway and deactivated FoxO1 in a time-dependent manner Inactivated FoxO1 reduced TLR4 expression and inflammation PI3K/Akt and FoxO1 are TLR4-regulated Established a self-limiting system of inflammation	[233]
Mixture Resveratrol and Curcumin	(Phenolic compounds)	Effects in IL-1β-induced chondrocytes: Anti-inflammatory, antiapoptotic and anti-cytotoxic synergistic effects Increased antiapoptotic proteins Bcl-2, Bcl-xL and Traf1 in a time-dependent manner Supressed NF-κB activation and nuclear translocation in a time- and concentration-dependent manner Inhibited COX-2, MMP-3, MMP-9, VEGF, caspase-3, and PARP cleavage levels Increased COL2A1 and SOX-9 production Resveratrol blocked IκBα degradation and curcumin inhibited IKK	[234]
		Effects in IL-1β- or U0126-stimulated chondrocytes: Showed synergistic chondroprotective efficacy and ameliorated inflammatory effects Decreased apoptotic cells and resveratrol potentiated antiapoptotic effects of curcumin Inhibited caspase-3 activation and degradation of β-integrins Blocked the downregulation of Erk1/2 in a dose- and time-dependent manner	[174]
Sanguinarine	The roots of: *Sanguinaria canadensis* Benzophenanthridine alkaloid	Effects in IL-1β-induced chondrocytes: Inhibited OA progression Inhibited *MMP1a, MMP3, MMP13*, and *ADAMTS5* mRNA and protein expression Inhibited NF-κB and JNK signalling pathways	[235]
Schisantherin A	The fruits of: *Schisandra sphenathera* Dibenzocyclooctadiene Lignan	Effects in IL-1β-induced chondrocytes: Anti-inflammatory and chondroprotective Inhibited NOS2, COX-2, NO, PGE2, and TNF-α, MMP-1, MMP-3, and MMP-13 production Inhibited NF-κB p65 translocation to the nucleus, and inhibited MAPKs activation and IκBα degradation in a dose-dependent manner	[236]
Sesamin	*Sesamun indicum* sesame seed oil lignan	Effects in IL-1β induced chondrocytes: Inhibited p38 and JNK phosphorylation Decreased *MMP1, MMP3* and *MMP13* mRNA and protein expression	[237]
Sulforaphane	*Brassica oleracea italica* cruciferous vegetables (abundant in broccoli) Isothiocyanate	Effects in IL-1β- or TNF-α-treated OA chondrocytes/cartilage explant: Showed anti-inflammatory and immune-modulatory effects Induced the phase 2 enzymes activity NQO1 (one of the most potent inducers) Inhibited NF-κB p65 pathway by down-regulating IκB-α degradation and IKK-αβ and IκB-α phosphorylation Inhibited *COX2, PTGES* and *NOS2* mRNA and protein expression even at low concentrations Inhibited PGE2 and NO production in chondrocytes and explant culture Suppressed PG and COL2A1 degradation in cartilage explant culture	[238]
		Effects in IL-1 or TNF-α-treated OA chondrocytes: Sulforaphane was not cytotoxic at up to 20 μM Demonstrated anti-inflammatory mechanism mediated by NQO1 activity Inhibited NF-κB and JNK activation Inhibited *MMP1, MMP3* and *MMP13* mRNA and protein expression	[239]
		Effects in C-28/I2 cell line/OA chondrocytes induced by TNF/CHX, DENSPM/CHX, H_2_O_2_ GROα: Showed cytoprotective effects Inhibited apoptosis, hypertrophic differentiation and ECM degradation Reduced the active/phosphorylated JNK Inhibition of p38 MAPK phosphorylation and suppressed caspase 3, caspase 8 and caspase 9 activation Increased active/phosphorylated Akt protein	[240]
		Effects in IL-1/OSM-induced OA chondrocytes/SW-1353 cell line/synovial cells: Inhibited *ADAMTS4, ADAMTS5, MMP1, MMP13*, and mRNA expression (sulforaphane acted independently of NRF2) in chondrocytes and synoviocytes Induced *HMOX1* (an NRF2-regulated gene) mRNA expression Inhibited *NOS2*, *IL6*, *IL8* genes Blocked inflammation and inhibited cartilage destruction by attenuating NF-κB signalling Inhibition of p38 MAPK isoform Accumulated sulforaphane-GSH metabolites	[241]
		Effects in knee OA patients: Isothiocyanates were detected in the synovial fluid and in blood plasma of the high glucosinolate group, but not the low one Demonstrated biological impact on the joint tissues Synovial fluid protein profile and common plasma proteins showed significantly different levels of expression between both groups Decreased *CXCL10* and increased *IRX3* in fat tissue in the high-glucosinolate group	[242]
Sulforaphane– microsphere system	Sulforaphane-Poly (D, L-lactic-co-glycolic) acid (PLGA) microspheres	Effects in LPS-induced OA chondrocytes: Showed chondroprotective properties Inhibited anti-inflammatory markers Inhibited *COX2, ADAMTS5* and *MMP2* mRNA and protein expression	[243]
Taraxasterol	*Taraxacum officinale* Pentacyclic-triterpene	Effects in IL-1β-stimulated chondrocytes: Suppressed inflammatory mediators via inhibition of NF-κB p65 translocation from cytoplasm to nucleus and IκBα degradation Inhibited NO, NOS2, PGE2, COX-2, MMP-1, MMP-3, and MMP-13 production in a dose-dependent manner	[244]
Terpenoid compounds (tuberatoide B, loliolide, sargachromenol, sargachromanol D, sargachromanol G, sargaquinoic acid, sargahydroquinoic acid, isoketocharolic acid/IKCA, isonahocol E3, and fucosterol) Phlorotannins Eicosapentaenoic acid EPA	*Sargassum seaweed* (Terpenoids) Polyphenols Fatty acid	Effects in IL-1β-induced SW1353 cell line: Inhibited oxidative stress and inflammatory responses Suppressed NF-κB, p38 MAPK, and PI3K/Akt signalling pathways Inhibited IL-1β-induced *NOS2* and *COX2* mRNA and protein expression Decreased NO and PGE2 production Inhibited IL-1β-induced *MMP1, MMP3*, and *MMP13* mRNA and protein expression	[245]
Thymoquinone (active metabolite)	*Nigella sativa* Black cumin oil Monoterpene	Effects in IL-1β-stimulated OA chondrocytes: Showed chondroprotective and anti-inflammatory effects via inhibition of NF-κB p65 and MAPKs activation Inhibited IκBα degradation Suppressed COX-2, NOS2, NO, PGE2, MMP-1, MMP-3, and MMP-13 production	[246]
Wogonin	The root extract of: *Scutellaria baicalensis* Flavone	Effects in IL-1β-induced OA chondrocytes: Showed chondroprotective effects Decreased *IL6* and *MMP13* mRNA and protein expression in a dose-dependent manner Suppressed *MMP3, MMP9* and *ADAMTS4* mRNA expression Suppressed oxidative and nitrosative stress by suppressing *NOS2* gene and protein expression, ROS and reactive nitrogen species Supressed *COX2* mRNA and protein expression and PGE2 production Inhibited c-Fos/AP-1 activity Enhanced *COL2A1* and *ACAN* gene expression	[247]
		Effects in IL-1β-induced OA cartilage explant: Suppressed glycosaminoglycan release Effects in IL-1β-induced OA chondrocytes: Suppressed oxidative stress, inflammation and matrix degradation Increased NRF2 activation and activated transcription of *NRF2*-dependent genes *HO1, GCLC, SOD2* and *NQO1* and the upstream kinase ERK1/2 Inhibited *MMP13, MMP3, MMP9, ADAMTS4* mRNA expression and protein expression Inhibited *IL6, COX2* and *NOS2* mRNA and protein expression Inhibited NO and PGE2 production Upregulated *COL2A1,* and *ACAN* mRNA and protein expression Effects in IL-1β-induced cartilage explants: Restored COL2A1 and GAG contents in a dose-dependent manner	[248]
		Effects in IL-1β-induced OA chondrocytes: Demonstrated cytoprotective properties Showed genomic DNA binding ability through intercalation mechanism, and the intercalation was found between DNA base pairs guanine and cytosine Inhibited genomic DNA fragmentation and ROS generation Provided stability of DNA against chemical denaturation Inhibited DNA denaturation mediated by dimethylsulphoxide (DMSO) Inhibited apoptosis and apoptotic pathways and upregulated antiapoptotic proteins	[249]

**Table 2 pharmaceuticals-17-01148-t002:** Bioactive compounds and nutraceuticals for the management, treatment, or prevention of OA in animals.

Bioactive Compounds	Sources/Classes	Effects of Bioactive Compounds	Ref.
ALM16 Herbal mixture Major active compounds: (calycosin, calycosin-7-O-β-D-glucopyranoside) lithospermic acid	Dried roots of (*Astragalus* *membranaceus*) Isoflavonoids (*Lithospermum* *erythrorhizon*) Phenolic acid	Effects in OA cartilage/OA-induced rats: Showed synergistic or additive chondroprotective properties of each extract Demonstrated a potent protective effect on articular cartilage, anti-inflammatory and analgesic actions (dose 200 mg/Kg) Attenuated histopathological lesions in cartilage, pain symptoms, mechanical allodynia, and thickness of the paw edema	[146]
Amurensin H (Vam3)	*Vitis amurensis* Dihydroxy-stilbene Oligostilbenoid (resveratrol dimer)	Effects in IL-1β-stimulated rat chondrocytes: Showed anti-inflammatory and chondroprotective effects Inhibited oxidative stress, mitochondrial damage and ECM degradation (increased glycosaminoglycan and Col2a1 levels) Inhibited Nos2, nitric oxide, Pge2, Cox-2, Il-6, Il-17, Tnf-α, Mmp-9, Mmp-13 levels, Tlr4, Traf-6, Syk and Nf-κb protein expression in a dose-dependent manner Effects in OA cartilage/subchondral bone: decreased OA progression, cartilage fibrillation, cartilage loss, subchondral bone erosion and inflammation	[250]
Arctigenin (Phenylpropanoid dibenzylbutyrolactone)	*Arctium lappa* Greater burdock Lignan	Effects in OA cartilage Inhibited OA development, attenuated histological damage and showed lower OARSI score Mitigated cartilage erosion, hypocellularity and PG loss	[152]
Artesunate (Artemisinin)	*Artemissia annua* Sesquiterpene lactone	Effects in osteoclast/synovium/OA-induced rat: Showed anti-inflammatory activity Inhibited osteoclastogenesis and angiogenesis Downregulated Vegf, Hgf and Angp1 Inhibited Il-6, Il-1β, Tnf-α, Pge2 activity and JAK/STAT pathway Increased Col2a1, Il-4, Igf-1 and Tgf-β	[251]
		Effects in rat OA cartilage: Inhibited OA development Upregulated Igf-1 and reduced Opn and c-telopeptides of type II collagen levels	[252]
Avocado/soybean Unsaponificables ASU (β-sitosterol, campesterol, and stigmasterol) Triterpenes	*Persea gratissima* and *Glycine max* mixture of avocado and soybean unsaponifiables (Phytosterols) Triterpene alcohols	Effects in bovine articular chondrocytes: Showed chondroprotective properties Enhanced *Tgfb1*, *Tgfb2* mRNA expression Increased Pai-1 production Induced ECM repair mechanisms	[253]
		Effects in bovine chondrocytes: Showed anti-inflammatory effects Reduced the progression of cartilage damage Inhibited *Tnfa, Il1b, Cox2*, and *Nos2* gene expression and downregulated Pge2 and nitrite production in LPS-activated chondrocytes	[159]
		Effects in OA cartilage/synovial membrane/subchondral bone/OA-induced rat: Showed anti-oxidative and anti-inflammatory properties in MIA-induced OA rat Reduced histopathological damage of all joint tissues with a significant decrease in the Mankin score Decreased Tnf-α and Mmp-13 and increased Col2a1 and Acan synthesis Reduced Nos2 in both OA cartilage and subchondral bone	[254]
Mixture: ASU and Epigallocatechin-3-O-gallate		Effects in IL-1β and TNF-α-activated equine chondrocytes: This combination potentiated the anti-inflammatory activity Suppressed *Cox2* gene expression and Pge2 production, related to Nf-κb translocation inhibition from cytoplasm to the nucleus	[255]
		Effects in equine chondrocytes: Demonstrated anti-inflammatory activity in cytokine-activated articular chondrocytes Decreased *Tnfa, Il6, Cox2* and *Il8* gene expression and Pge-2 synthesis through Nf-κb nuclear translocation inhibition	[256]
ASU + α-lipoic acid combination		Effects in LPS, IL-1β or H_2_O_2_-activated equine chondrocytes: Showed a potential combination of anti-inflammatory and antioxidant capacities in OA management Inhibited Pge-2 production significantly more than ASU alone or α-lipoic acid alone Reduced nuclear translocation/activation of Nf-κb	[257]
Combination (ASU +glucosamine +chondroitin)		Effects in canine chondrocytes: The combination stimulated the anti-inflammatory effect of a low concentration of NSAID for OA management Stronger inhibitory effect on Il-6, Il-8, and Mcp-1 production than carprofen in IL-1β-stimulated chondrocyte microcarrier spinner cultures The combination together with a lower dose of carprofen reduced Pge2 production significantly more than either treatment alone	[258]
Baicalin	(*Scutellaria baicalensis* *Georgi*) Mainly extracted from dry root Flavone glycoside (flavonoid)	Effects in mice OA cartilage/synovium/OA-induced mice: Attenuated OA progression Decreased PG loss, cartilage degradation and the OARSI scores Ameliorated synovitis	[157]
		Effects in mouse chondrocytes: Enhanced ECM synthesis by activating the Hif-1α/Sox-9 pathway and chondrogenic marker expression Increased *Col2a* and *Acan* gene expression Inhibited catabolic genes: *Adamts5*, *Mmp9*, *Mmp13* and prolyl hydroxylases	[259]
		Effects in rat chondrocytes: Inhibited oxidative activity, ROS production and apoptotic cell death of endplate chondrocytes induced by H_2_O_2_ Upregulated *Enos* mRNA Reduced malondialdehyde levels and increased sod Downregulated apoptotic signalling indicators: Parp cleavage, Bax and pro-Casp-3 protein expression	[260]
Berberine	Medicinal herbs: *Hydrastis canadensis* *Berberis aristate* *Cortex phellodendri* *Coptis chinensis* isoquinoline-derivative alkaloid	Effects in IL-1β-induced rabbit chondrocytes: Inhibited *Mmp3* and *Adamts5* gene expression in chondrocytes Increased *Timp1, Acan* and *Col2a1* gene expression Effects in rabbit cartilage explants: Inhibited cartilage degradation Inhibited release of collagen and GAG fragment	[261]
		Effects in IL-1β-induced rat chondrocytes/cartilage explants: Showed chondroprotective properties and reduced articular cartilage destruction Inhibited glycosaminoglycan release and no production of high-dose berberine Suppressed *Mmp1, Mmp3* and *Mmp13* mRNA and protein expression in a dose-dependent manner and upregulated *Timp1* mRNA and protein expression in chondrocytes/cartilage explant (100 µm optimum concentration)	[262]
		Effects in IL-1β-stimulated rat chondrocytes: Showed the maintenance of chondrocyte survival and promoted matrix production in IL-1β-stimulated articular chondrocytes Activated Akt/p70S6K/S6 signalling pathway Effects in rat OA cartilage: Protected articular cartilage and reduced matrix degradation Enhanced Col2a1, p-Akt and p-S6 levels	[263]
		Effects in rat chondrocytes: Attenuated SNP-stimulated chondrocyte apoptosis via activating AMPK signalling and inhibition of p38 MAPK activity Suppressed SNP-induced Nos2 protein expression Effects in OA cartilage: Showed chondroprotective effect Decreased cartilage degradation, Casp-3, and Bax protein expression Increased Bcl-2 expression, and enhanced Col2a1 synthesis	[264]
		Effects in rat chondrocytes: Promoted SNP-stimulated chondrocyte proliferation via activation of Wnt/β-catenin pathway Upregulated *Ccnd1, Ctnnb1* and *Myc* gene expression Reduced *Gsk3b* and *Mmp7* mRNA expression Effects in OA cartilage: Decreased OA progression and cartilage degradation Reduced Mankin scores Enhanced Ctnnb1 and Pcna expression	[265]
		Effects in IL-1β -induced rat OA cartilage: Prevented cartilage degradation Inhibited PG loss Decreased immunostaining of IL-1β in the superficial and middle zones of cartilage	[158]
		Effects in rat chondrocytes: Demonstrated anti-catabolic and anti-inflammatory properties Inhibited *Nos2, Cox2, Mmp3, Mmp13, Tnfa,* and *Il6* mRNA and protein expression Decreased the phosphorylation of MAPK (ERK, JNK, and p38) signalling pathway Increased Col2a1 protein expression	[266]
Butein	*Rhus verniciflua* stem bark of cashews and the genera Dahlia, Butea, Searsia (Rhus) and Coreopsis are common sources Chalcones (flavonoids)	Effects in rat OA cartilage/synovium/subchondral bone: Inhibited PG loss and cartilage fibrillation and degradation Decreased OARSI score Alleviated synovitis Reduced subchondral bone plate thickness	[159]
Celastrol	(*Tripterygium wilfordii* *Hook F.*) root bark “Thunder of God Vine” Pentaciclic Triterpenes	Effects in rat chondrocytes/OA articular cartilage (dose-dependent manner): Inhibited inflammatory response and Nf-κb signalling pathway Ameliorated apoptosis by enhancing autophagy Decreased cleaved Casp-3, p-IκBα, p-p65 protein expression and *Bax, Sqstm1, Il6, Tnfa* mRNA and protein expression Increased *Bcl2, Ccnd1* mRNA and protein expression and Lc3-II levels Attenuated articular cartilage degradation Ameliorated cartilage loss and osteophyte formation	[267]
		Effects in OA cartilage: Attenuated cartilage damage and joint pain Suppressed *Sdf1/Cxcr4* mRNA pathway Decreased *Mmp13* and *Adamts5* mRNA and protein expression Increased *Col2a1* and *Acan* mRNA expression	[268]
		Effects in rabbit chondrocytes: Decreased apoptosis via Atf6/Chop pathway Inhibited *Bip, Aft6, Chop* and *Xbp1* (endoplasmic reticulum stress, ERs markers) mRNA and protein expression Decreased *Casp3* and *Casp9* mRNA and protein expression Effects in rat OA articular cartilage/synovium: Reduced cartilage injury, synovial hyperplasia and wear in the knee joints	[269]
Celastrol Nanocomplex	Celastrol+ Hollow mesoporous silica nanoparticles+Chitosan	Effects in rat chondrocytes Inhibited Mmp-3, Mmp-13, Il-1β, Tnf-α levels and Nf-kb signalling pathway Reduced inflammation Effects in OA cartilage/synovium/subchondral bone/OA-induced rat: Demonstrated high biosolubility and decreased cartilage damage Showed protective effect on cartilage and subchondral bone Reduced knee swelling and synovial inflammation	[270]
Compound K	*Panax ginseng* roots, fruits, leaves, flower buds Gingenoside (tetracyclic triterpenoid)	Effects in mouse pre-osteoblastic MC3T3-E1 cells: Protected against H_2_O_2_-induced cytotoxicity Alleviated inflammatory response Stimulated osteoblastic cell differentiation and mineralization Inhibited ROS and NO levels Increased *Alp, Col2a,* and *Ocn* mRNA Decreased *Ikk* and *Il1b* mRNA expression	[271]
Criptotanshinon	(*Salvia miltiorrhiza* *Bunge*) Extracted from the root of the plant Diterpene quinones	Effects in OA cartilage/suchondral bone/OA-induced mice: Decreased cartilage destruction and protected against OA progression OARSI scores and subchondral bone plate thickness reduction	[163]
Crocin		Effects in mouse skeletal muscle cell line C2C12: Suppressed Il-6 by downregulation of Jnk level Effects in muscle tissue/OA-induced rats: Reduced joint pain, inflammation, muscular lipid peroxidation and *Nrf2* mRNA expression Attenuated muscular oxidative stress through inhibiting muscular ROS generation Attenuated muscle dysfunction and decreased muscular Il-6 production Increased citrate synthase activity and *Myh9* mRNA expression Increased glutathione production and *Gpx1* mRNA and activity	[272]
		Effects in IL-1β-induced rabbit chondrocytes: Inhibited *Mmp1, Mmp3* and *Mmp13* gene and protein expression Inhibited Nf-κb pathway and suppressed degradation of IκBα Effects in rabbit OA cartilage: Suppressed cartilage degradation Reduced *Mmp1, Mmp3* and *Mmp13* genes	[273]
Curcuminoids: Curcumin Demethoxycurcumin, Bisdemethoxycurcumin	(*Curcuma longa*) (*Curcuma domestica*) Turmeric rhizome Diarylheptanoids (Phenolic compounds)	Effects in IL-1β-stimulated equine articular cartilage explants: Inhibited cartilage degradation Decreased GAG release at high concentrations	[274]
		Effects in IL-1β-stimulated equine cartilage explants: Showed anti-catabolic and anti-inflammatory properties at low concentrations (non-cytotoxic concentrations) Reduced PG loss Decreased Pge2 and Mmp-3 release	[275]
		Effects in rat temporomandibular joint OA cartilage: Showed anti-inflammatory and chondroprotective properties Reduced cartilage erosion and PG loss Decreased *Nos2, Cox2, Il1b, Mmp9, Mmp13* protein levels and increased Nrf2 protein level	[172]
		Effects in IL-1β-induced rat chondrocytes: Blocked Nf-κb signalling pathway by suppressing *Ikba* mRNA phosphorylation and subunit *Rela* mRNA nuclear translocation Decreased *Mmp13* mRNA and protein expression, and upregulated *Col2a1* mRNA and protein expression in a time-dependent manner	[276]
		Effects in IL-1β-induced rat chondrocytes: Suppressed apoptosis marker (Casp-3) through autophagy via Mapk/Erk1/2 activation pathway and increased autophagy markers (Lc3-II, and Beclin-1)	[277]
		Effects in rats OA cartilage/synovial tissues/rat OA-induced knee: Improved inflammatory lesions by intra-articular injection Inhibited LPS-induced overexpression of *Tlr4* and its downstream *Nfkb* pathway mRNA and protein expression Decreased inflammatory cytokines LPS-induced Il-1β and Tnf-α production in synovial membrane	[278]
Curcumin nanoparticles	Topical treatment	Effects in cartilage/OA mice: Slowed OA progression and decreased ECM degradation, cartilage erosion, and aggrecan loss Reduced Mmp-13 and Adamts-5 levels Reduced pain and improved locomotor behaviour Effects in infrapatellar fat pad: Suppressed *Cfd*, *Lep*, *Adipoq*, adipo-regulatory transcription factors/enhancer binding protein alpha and peroxisome proliferator-activated receptor gamma, and *Mmp13* and *Adamts5* mRNA Effects in synovium/subchondral bone: Reduced synovitis and subchondral plate thickness	[173]
Mixture: Curcuminoids Hydrolysed collagen and Epigallocatechin-3-gallate	(*Curcuma longa* L.) Turmeric Polyphenols Hydrolysed collagen (High levels of glycine and proline, amino acids essential for stability and cartilage regeneration) (*Camellia sinensis*) Green tea Epigallocatechin-3-gallate (Flavanol)	Effects in IL-1β-stimulated bovine chondrocytes: Demonstrated anticatabolic, anti-inflammatory, additive and synergistic properties Decreased *Il6, Nos2, Cox2, Mmp3, Adamts5* and *Adamts4* gene expression Inhibited NO, Pge2 production	[180]
Herbal composition LI73014F2 (2:1:2 ratio): Gallic acid, chebulagic acid, chebulic acid, chebulinic acid, gallotannins, ellagitannins (punicalagin), ellagic acid Diferuloylmethane Demethoxycurcumin Bisdemethoxycurcumin, and turmeric acid Boswellic acids: 3-O-acetyl-11-keto-β-boswellic acid, 11-keto-β-boswellic acid, and β-boswellic acid	(*Terminalia chebula*) Fruit myrobalan Tannins (polyphenols) (*Curcuma longa*) Polyphenols (*Boswellia serrata*) Olibanum Pentacyclic triterpenes	Effects in cartilage/synovium/OA-induced rats: Decreased pro-inflammatory mediators such as Cox-2, Pge2, Lox5, and Ltb-4 Decreased pro-inflammatory cytokines: Il-1β, Il-6, and Tnf-α, 89%, 84%, and 38%, respectively Reduced Mmp-2, Mmp-3, Mmp-13 levels Alleviated joint pain by suppressing synovial membrane and cartilage degradation (dose 50 mg/Kg/day for 3 weeks)	[279]
Ellagic acid	Fruit peel of raspberries, strawberries, cranberries, pomegranate, walnuts, pecans, grapes Dimeric derivative of gallic acid Phenolic compound	Effects in cartilage/synovium/OA-induced mouse Protected against cartilage degradation Inhibited PG loss Decreased OARSI score Alleviated synovitis Delayed OA progression	[186]
Emodin	Root and rhizome of *Rheum palmatum* Anthraquinone derivative (Phenols)	Effects in IL-1β-induced rat chondrocytes: Decreased *Mmp3, Mmp13, Adamts4* and *Adamts5* mRNA and protein expression by suppression of NF-κB and Wnt/β-catenin pathways Increased *Acan* and *Col2a1* mRNA and protein expression Effects in cartilage/OA-induced rats: Protected against the development and progression of OA Reduced cartilage degradation Decreased *Mmp3, Mmp13* and *Ctnnb1* mRNA	[280]
		Effects in IL-1β-induced rat chondrocytes: Reduced cytotoxicity in a dose-dependent manner Inhibited no and pge-2 levels, and *Mmp1* and *Mmp13* mRNA expression Inhibited ERK activation and Wnt/β-catenin pathway	[281]
		Effects in IL-1β-induced rat chondrocytes/cartilage: Alleviated inflammation and reduced *Mmp3, Mmp13* and *Adamts4* mRNA and protein expression Reduced cartilage matrix degradation Protected knee joint cartilage Effects in serum/OA-induced rat: Inhibited Nos2, No, Cox-2 and Pge2 levels Emodin at 80 mg/Kg is comparable to celecoxib at 2.86 mg/Kg	[282]
Fatty acids n-3 PUFAs omega 3 polyunsaturated fatty acids	Soybean, canola, olive oils, flaxseed, walnuts, marine phytoplankton and fish oil ALA: α-linolenic acid EPA: eicosapentaenoic DHA: docosahexaenoic	Effects in equine synoviocyte culture: n-3 PUFAs EPA and DHA modulated inflammatory response and reduced *Adamts4, Mmp1, Mmp13, Il1b, Il6,* and *Cox2* genes, stimulated by recombinant equine (re) IL-1β DHA-derived docosanoids such as resolvin D1 and D2, maresin 1 and protectin DX reduced *Adamts4, Mmp1, Mmp13, Il6,* and *Cox2* genes	[283]
		Effects in IL-1β-mediated bovine cartilage explants: EPA and DHA reduced ECM degradation Demonstrated that EPA maintained a reduced expression of *Adamt4, Adamts5, Mmp3* and *Mmp13,* and *Cox2* gene until the end of the 5-day treatment	[284]
		Effects in IL-1α-induced bovine chondrocytes: n-3 PUFAs showed favourable effects against inflammation and cartilage degradation EPA was the most effective, then DHA and ALA n-6 PUFA, arachidonic acid had no effect n-3 PUFAs reduced *Cox2, Adamts4, Adamts5, Mmp3, Mmp13, Il1a, Il1b* and *Tnfa* mRNA	[285]
		Effects in OA cartilage/OA-induced mouse EPA intra-articular injection treatment decreased matrix degradation and Mankin scores Reduced Mmp-13 protein expression Inhibited OA progression	[194]
Geniposide	Extract of the fruit *Gardenia jasminoides* *Ellis,* zhizi Iridoid glycoside (monoterpenoids)	Effects in rabbit OA chondrocytes/synovial fluid/OA-induced rabbit: Showed anti-inflammatory effects by inhibiting p38 MAPK signalling pathway Inhibited *Il1b, Tnfa,* and *Mmp13* gene expression and protein expression Inhibited oxidative stress	[286]
		Effects in IL-1β-induced rat chondrocytes: Inhibited inflammation and apoptosis Inhibited Bax, Cyto-c, cleaved-Casp3, no, Pge2, Nos2, Cox-2, and Mmp-13 protein expression Increased Bcl-2 and Col2a1 protein expression Inhibited Pi3k/Akt/Nf-κb phosphorylation signalling pathway Effects in OA cartilage/OA-induced rat: Reduced cartilage damage and OARSI scores Inhibited OA progression	[287]
		Effects in rat chondrocytes: Promoted chondrocytes proliferation Inhibited sodium nitroprusside-induced apoptosis via the reduction of NO levels	[288]
Genistein	(*Gycine max*) soybean Isoflavone (flavonoids)	Effects in OA condyle cartilage/temporomandibular joint in OA-induced rat: Observed more therapeutic effects on cartilage repair at high doses Decreased NF-κB phospho-p65 signalling Inhibited *Il1b* and *Tnfa* mRNA expression	[289]
		Effects in IL-1β-induced OA cartilage/OA-induced rat Reduced inflammation and prevented ECM degradation Decreased OARSI score and attenuated OA progression	[196]
		Effects in IL-1β-induced OA cartilage/OA-induced rat Reduced cartilage degradation Increased collagen II, Acan, and ERα levels Downregulated caspase 3 levels Effects in synovial fluid: Reduced Tnf-α and Il-1β levels	[197]
Halofuginone	*Dichroa febrifuga* Alkaloid	Effects in cartilage/OA-induced rodents: Decreased PG loss and articular cartilage calcification Reduced Col10, Mmp-13 and Adamts-5 Increased lubricin, Col2a1, and Acan levels Effects in subchondral bone: Inhibited osteoclastogenesis by decreasing Th17 cells and Rankl expression Inhibited osteoid islets’ formation by suppressing Tgf-β activity Attenuated aberrant angiogenesis	[290]
		Effects in cartilage/OA-induced mice Attenuated cartilage degradation and OA progression Reduced Col10 and Mmp-13 levels Effects in subchondral bone: Improved subchondral bone microarchitecture Reduced abnormal bone resorption Decreased abnormally elevated Tgf-β activity and release from bone mineral matrix and inhibited osteoid islets’ formation Inhibited aberrant angiogenesisin in early-stage OA administered by oral gavage	[291]
		Effects in ATDC5 murine chondrogenic cell line: 6.25-25 ng/mL did not affect chondrocytic viability Inhibited Tgf-β1 signalling and downregulated p-Smad2 protein in a dose- and time-dependent manner Effects in cartilage/OA-induced murine: Prevented cartilage damage by Tgf-β1 signalling inhibition Reduced p-Smad2/3 levels Downregulated PG loss Decreased *Col10* expression and Mmp-13 levels	[292]
Harpargoside, harpagide and procumbide β-cariofileno, α-humuleno and α-copaeno Oleanolic acid, ursolic acid and 3β-acetyloleanolic acid Eugenol Acteoside and isoacteoside	*Harpagophytum procumbens* (HP) Devil’s claw root extract Iridoid glucosides Sesquiterpenes Triterpenes Monoterpene Phenolic glycosides	Effects in cartilage/OA-induced rabbit: Showed chondroid regeneration Increased elastic and collagen fibres Increased *Timp2* mRNA expression	[293]
Hydroxytyrosol (HT)	*Olea europea* L. Olive leaf extract Fruits Extra virgin olive oil HT is more abundant in processed fruit and olive oil Secoiridoid derivative	Effects in cartilage/synovial membrane/OA-induced rat Showed anti-inflammatory activity and prevented articular cartilage and bone destruction induced by kaolin and carrageenan Attenuated synovial membrane and periarticular soft tissue edema and reduced inflammatory infiltration Ameliorated paw swelling	[294]
		Effects in cartilage/synovial cells/STR/ort mice: Inhibited cartilage destruction and suppressed OA progression on knee joint Enhanced *Has2* mRNA expression and improved high molecular hyaluronan production by synovial cells	[295]
Hydroxytyrosol/Procyanidins (Oleogrape^®^SEED)	(Extract from olive and grape seed): (*Olea europea* L.) mainly found in olive leaf and oil Phenolic compound (*Vitis vinifera*, grape) Flavonoids Other sources: pine bark, cocoa, raspberry, vegetables, legumes, nuts	Effects in IL-1β-induced OA chondrocytes/OA-induced rabbit: Showed anti-inflammatory and chondroprotective properties Inhibited *Nos2, Cox2, Mmp13* genes and NO, Pge2 and Mmp-13 production Effects in cartilage: Reduced OARSI score and cartilage degradation Effects in serum: Downregulated NO, Pge2 and Mmp-13 levels Conserved their bioactivity and bioavailability in serum after undergoing digestive process	[296]
Hyperoside	(*Hypericum perforatum*) fruits and herbs of different plant families (Hypericaceae, Rosaceae, Ericaceae, Campanulaceae, and Labiatae) Flavonoid glycoside	Effects in IL-1β-induced chondrocytes/OA-induced mice: Inhibited inflammation and ECM degradation Reduced Nos2, Cox-2, Adamts-5, Mmp-3, and Mmp-13 Upregulated collagen II, Acan, and Sox-9 Suppressed Pi3k/Akt/Nf-κb and Mapk pathways Attenuated oxidative stress and apoptosis via Nrf2/Bax/Bcl-xl axis Decreased ROS levels Enhancing Nrf2/Ho-1 pathway to counteract Nf-κb activation Effects in cartilage: Inhibited GAG loss and cartilage degradation, and decreased the OARSI scores Increased Nrf2 levels	[297]
Icariin	*Epimedium sagittatum* flavonol glycoside	Effects in bone mesenchymal stem cells: Icarin promoted chondrogenic differentiation and Acan, Bmp2 and Col2a1 protein expression Effects in rabbit cartilage tissue: Repaired knee cartilage damage and enhanced Col2a1 expression (treatment with icarin plus bone mesenchymal stem cells was even more effective than the effect produced by either treatment alone in a time-dependent manner)	[298]
		Effects in ATDC5 cell line/rat chondrocytes: Promoted ECM secretion and enhanced *Col2a1* and *Sox9* gene expression in a concentration-dependent manner Enhanced *Ift88* gene and protein expression and ciliary assembly and promoted Erk phosphorylation Effects in cartilage/OA-induced rat: Improved histological cartilage phenotype and attenuated cartilage degradation	[299]
		Effects in TDP-43 chondrocyte lines/synovial tissue/serum/OA-induced rat Inhibited Tdp43 overexpression-induced apoptosis Attenuated the formation of neovascularization in the synovial tissue of a rat OA model Decreased Vegf and Hif-1α in synovial tissue and serum	[300]
		Effects in IL-1β-induced rat chondrocytes: Inhibited chondrocyte apoptosis and inflammatory cytokines’ production through the suppression of Nf-κb p65 phosphorylation and Mapk signalling Upregulated Akt activation Increased Ikbα protein Induced chondrocyte autophagy Decreased *Il6* and *Tnfa* gene and protein expression	[301]
		Effects in oxygen, glucose and serum deprivation-induced rabbit bone marrow-derived mesenchymal stem cells: Inhibited ERs markers levels and autophagy Protected against cytotoxicity and apoptosis by inactivation of Mapk signalling via three specific siRNAs (*Erk, p38* and *Jnk*) pathway	[302]
Indole tetracyclic alkaloids Oxindole alkaloids Indole pentacyclic alkaloid Glycoindole alkaloids Quinovic acids Tannins	*Uncaria guianensis* *Uncaria tomentosa* Cat’s claw Alkaloids Triterpenes heterosides Polyphenols	Effects in LPS-induced murine macrophages (RAW 264.7 cells): Showed antioxidants and anti-inflammatory Properties, potentially an effective treatment for OA-Inhibited Tnf-α and Pge2 production	[214]
Isofraxidin	*Siberian ginseng* and *Apium graveolens* Coumarin (phenolic compound)	Effects in OA cartilage/serum/OA-induced mouse: Reduced subchondral bone plate thickness and prevented calcification and erosion of cartilage Inhibited inflammatory cytokines in serum	[215]
Licochalcone A	*Glycyrrhiza glabra*, licorice root *Glycyrrhiza inflate* Flavonoids	Effects in IL-1β-induced rat chondrocytes: Reduced *Adamts5, Adamts4, Mmp13* and *Mmp1* mRNA expression Inhibited Ikkα/β and p65 phosphorylation, and increased Iκbα expression Inhibited Wnt/β-catenin signalling pathway Upregulated Col2a1 expression	[303]
		Effects in LPS-induced mouse chondrocyte: Mitigated ECM degradation by enhancing Acan and Col2a1production Decreased chondrocytes pyroptosis through Nrf2/Ho-1/Nf-κb pathway Inhibited *Nlrp3, Asc, Gsdmd, Casp1*, *Il18, Il1b* mRNA and protein expression Reduced Iκb-α degradation and the translocation of p65 Effects in cartilage/OA-induced mouse: Inhibited cartilage erosion and PG loss and reduced OARSI score Enhanced Nrf2 and mitigated OA progression Decreased Il-1β and Il-18 protein expression in air pouch mouse model	[304]
Ligustrazine (Tetramethylpyrazine)	*Ligusticum chuanxiong* *Hort* Rhizoma Alkaloids	Effects in IL-1β-exposed rat chondrocytes: Suppressed apoptosis and ER stress-related factors (Grp78 and Chop) Suppressed *Il6, Il1b, Nos2, Cox2, Tnfa, Mmp3, Mmp13, Adamts4* and *Adamts5* mRNA expression Prevented ECM destruction Increased *Acan* and *Col2a1* mRNA	[305]
Tetramethylpyrazine-Poly lactic-co-glycolic acid microspheres		Effects in cartilage/synovium/OA-induced rats: Improved efficacy and therapeutic effect by intra-articular injection of microspheres Demonstrated to be histologically safe Protected against cartilage damage Inhibited PG loss Decreased articular inflammation and reduced joint swelling	[306]
Magnoflorine	*Sinomenium acutum* alkaloid	Effects in subchondral trabecular bone/osteoblastic cell line/cartilage/OA-induced guinea pig: Promoted subchondral bone regeneration and prevented OA progression Stimulated osteoblasts’ proliferation and mineralization Upregulated *Lrp5, Ctnnb1, Runx2, Ocn* and *Erk2* mRNA expression and downregulated *Nfκb* (p105) gene in osteoblasts Attenuated cartilage degradation and increased Acan, Bmp7, Sox5, Tgf-β1 and chondrogenic cells	[307]
		Effects in cartilage/primary chondroprogenitor cells/synovial fluid/subchondral bone/OA-induced rats: Promoted cartilage regeneration and enhanced Acan, Bmp7, Sox5, Tgf-β1 and chondrogenic cells Increased chondrogenesis and chondrogenic signals such as *Col2a, Comp, Tnc* and *Sox9* mRNA expression and downregulated *Nf-κb* (*p105*) and *Erk2* gene in chondrogenic cells Decreased pro-inflammatory cytokines Il-17a, Il-12, Tnf-α, Inf-γ and Il-6 and increased anti-inflammatory cytokine Il-10 in synovial fluid Maintained the stabilization of trabecular bone microstructure	[308]
Myricetin	*Labisia pumila* *Trigonella foenum-graecum* L. Species of *Anacardium* and *Mangifera* (*Anacardiaceae*) Grapes, berries, chard spinach, broadbeans, garlic, peppers Flavonol	Effects in cartilage/OA-induced mice: Inhibited articular cartilage matrix degradation and reduced OARSI score by intragastric administration Inhibited inflammation response and ameliorated OA progression through Pi3k/Akt, which mediated the increased Nrf2/Ho-1 signalling pathway Inhibition of Pi3k/Akt signalling abolished Nrf2/Ho-1 pathway activation and the suppression of Nf-κb	[221]
Oleocanthal (decarboxymethyl ligstroside aglycone)	(*Olea europea* L.) Fruits, leaves, extra virgin oil Secoiridoid derivative (Phenolic compounds)	Effects of LPS-induced ATDC-5 murine chondrogenic cell line: Oleocanthal and its derivative 231 reduced Nos2 protein expression and NO production in a dose-dependent manner Decreased p38 protein expression at the highest dose (25 µM was linked to a cytotoxic effect) Synthetic derivative 231 showed no cytotoxicity even at higher concentrations	[309]
		Effects in LPS-induced murine chondrogenic cell line/murine macrophages: Demonstrated anti-inflammatory effects Inhibited *Mip1a* and *Il6* mRNA and protein expression in chondrocytes and macrophages Inhibited nitric oxide production via Nos2 downregulation and decreased Il-1β, Tnf-α and Gm-csf levels in macrophages	[310]
Procyanidin	(*Vitis vinifera*) grape seed extracts (*Malus pumila, Malus* *domestica Borkh. cv.* *Fuji*) Apple Procyanidins (flavonoid)	Effects in H_2_O_2_ or IL-1β-treated chondrocytes/cartilage/synovial tissue/OA-induced mice: Demonstrated anti-oxidant, antiapoptotic, and anti-inflammatory effects Enhanced *Acan* and *Col2a1* mRNA Suppression of *Nos2* mRNA expression Prevented heterotopic cartilage formation Reduced Inos protein levels in synovial tissues	[311]
		Effects in chondrocytes/OA-induced mice: Inhibited cartilage damage induced by mitochondrial dysfunction of chondrocytes Enhanced mitochondrial biogenesis with upregulation of *Pgc1a* gene expression Promoted mitochondrial dehydrogenase activity Upregulated *Acan* gene synthesis and regulated PG homeostasis Downregulated *Mmp3* and *Mmp13* catabolic genes	[312]
Puerarin	(*Radix puerariae*) Root of Pueraria Phytoestrogen (Isoflavone)	Effects in cartilage/OA-induced mice: Attenuated inflammatory responses Ameliorated cartilage damage and synovitis Effects in blood monocytes/macrophages: Decreased myeloid-derived C-C chemokine receptor 2+/lymphocyte Ag 6C+ monocytes Reduced *Ccl2* mRNA Suppressed proinflammatory monocyte recruitment	[226]
		Effects cartilage/OA-induced rats Anti-inflammatory and chondroprotective Ameliorated cartilage loss and upregulated Col2a1 levels Inhibited Mmp-3, Mmp-13, Adamts-5, and Cox-2 Effects in serum: Inhibited Il-1β, Il-6, and Tnf-α levels Inhibited OA biomarkers: Ctx-II, Ctx-I and Comp, stimulated the N-terminal propeptide of type II collagen expression, inhibited bone resorption and promoted bone formation	[313]
		Effects on IL-1β-induced chondrocytes: Suppressed inflammatory mediators, apoptosis, and ECM degradation by inhibiting Nf-κb through Nrf2 nucleus expression and activation and Ho-1 cytoplasm expression in a dose-dependent manner Decreased Bax and Casp-3 Reduced *Nos2, Cox2, Tnfa* and *Il6* mRNA and protein expression Decreased NO and Pge2 production Decreased Mmp-13 and Adamts-5 levels Upregulated Acan and Col2a1 Effects on cartilage/OA-induced mice: Decreased cartilage damage and OARSI score Alleviated OA progression and pain symptoms	[314]
		Effects on OA and OA-associated mitochondrial dysfunctions in rats: Alleviated mechanical hyperalgesia and cartilage damage Increased mitochondrial biogenesis Attenuated mitochondrial dysfunctions in OA rats AMPK inhibitor compound C abolished puerarin’s effects	[315]
Quercetin	*Achyranthes bidentata* *Ageratum conyzoides* *Chrysanthemum psyllium,* *Eleutherococcus senticosus* *Juglans regia* L. flowers, leaves, and fruits broccoli, onions, apples, berry crops, grapes, dark cherries, and green vegetables Flavonol (flavonoid)	Effects in cartilage/serum/synovial tissue/ synovial fluid/OA-induced rabbit: Showed comparable effects as celecoxib Reduced cartilage damage and OARSI score Inhibited Mmp-13, oxidative stress and increased Sod (major active molecule to scavenge free radical) and Timp-1 levels	[316]
		Effects in IL-1β-induced chondrocytes: Showed anti-inflammatory, antiapoptotic and immunomodulatory effects Inhibited the degradation of cartilage matrix, *Col2a1* and *Acan* mRNA and protein expression Inhibited Akt activation and Iκbα degradation Inhibited Nf-κb p65 phosphorylation and translocation into the nucleus Decreased Pge2, NO, and *Mmp13, Nos2* and *Cox2* mRNA expression and protein levels Decreased *Adamts4* mRNA expression Decreased apoptosis by inhibiting Casp-3 Restored mitochondrial membrane potential Effects on synovial macrophage/OA-induced rat: Induced M2 polarization of macrophages and promoted pro-chondrogenic cytokines for cartilage repair, and attenuated OA progression	[317]
		Effects in OA-induced rats: Showed anti-inflammatory effects and reduced toe volume and joint diameter Alleviated OA symptoms in a dose-dependent manner Effects in serum: Inhibited Il-1β and Tnf-α production Effects in joint tissues: Improved cartilage structure Suppressed Tlr4 and Nf-κb pathway	[318]
Quercetin Nanoparticle gel	Flavonol	Effects in blood serum/OA-induced rat Quercetin-loaded nanoparticle gel and *A. conyzoides* L. extract gel reduced Il-1β, Mmp-9, Mmp-13 and Adamts-5 levels Effects in knee joint: Prevented OA progression and PG degradation	[319]
Compound: Quercetin with palmitoylethanolamide (PEA-Q)	Flavonol with fatty acid amide	Effects in cartilage/OA-induced rat Reduced histological cartilage damage induced by sodium monoiodoacetate injection Decreased hyperalgesia and infiltration of inflammatory cells and reduced myeloperoxidase induced by carrageenan Improved locomotor function Effects in serum: Reduced Il-1β, Tnf-α, Mmp-1, Mmp-3 and Mmp-9, as well as nerve growth factor levels associated with nociceptive and neuropathic pain Showed similar or even greater effects when compared to oral meloxicam	[320]
Resveratrol	Root extracts of the weed *Polylygonum cuspidatum* *Vitis vinifera* red grapes, blueberries cranberries, peanuts, Stilbenes (polyphenols)	Effects in cartilage/OA-induced mice Reduced articular cartilage damage and Mankin and OARSI scores Decreased pro-inflammatory cytokine levels by Tlr4/Nf-κb signalling inhibition via downregulation of Myd88-dependent and -independent signalling pathways Activation of Pi3k/Akt pathway	[228]
		Effects in cartilage/OA-induced rabbit Exhibited cartilage-protective effect in a dose-dependent manner of 10–50 μMol/Kg Reduced matrix PG content loss Inhibited chondrocyte apoptosis *in vivo* Effects in synovial fluid: Reduced No production	[321]
		Effects in cartilage/OA-induced rabbits: Protected against cartilage destruction by intra-articular injection (10 µMol/Kg resveratrol once a day for two weeks) Decreased cartilage lesions such as fibrillation and fissures and reduced matrix PG content loss Effects in synovium: Statistically, scores of synovial inflammation did not show difference between control rabbits receiving DMSO only and resveratrol in DMSO groups	[322]
		Effects in joint tissues/OA-induced rats Tnf-α, Il-1β, Il-6, Il-18, Casp-3 and Casp-9 activity inhibition Suppressed Nf-κb and Nos2 protein expression Activated Ho-1/Nrf-2 signalling	[323]
		Effects in cartilage/OA-induced C57BL/6J mice fed a high-fat diet: Inhibited cartilage lesion and suppressed chondrocyte apoptosis on obesity-related OA Decreased body weight in obese mice and inhibited OA development by reducing biomechanical overloading and inflammatory factors (doses of 22.5 mg/Kg and 45 mg/Kg) by oral gavage Reduced the degradation of Col2a1 Effects in serum: Reduced triglyceride and cholesterol levels in serum but none these reductions were statistically significant Decreased levels of Ctx-II (45 mg/Kg doses)	[324]
Rutin (quercetin-3-O-rutinoside) Oleuropein Rutin/Curcumin	Abundantly found in: *Ruta graveolens*, rue Passionflower Buckwheat Apple Flavonol	Effects in cartilage/blood samples/synovium/OA-induced guinea pig: Decreased OA progression, reduced cartilage degradation and protected against inflammatory and catabolic processes Rutin decreased OA biomarkers: Coll2-1, Coll2-1NO2, and args neoepitope aggrecan fragments levels in serum Oleuropein decreased osteophyte formation in cartilage, decreased synovial histological score and decreased Pge2 and Coll2-1NO2 levels in serum Rutin/curcumin mixture decreased Coll2-1, Fib3-1 and Fib3-2 in serum	[325]
Sanguinarine	The roots of: *Sanguinaria canadensis* Benzophenanthridine alkaloid	Effects in IL-1β-induced cartilage explants: Inhibited OA progression and protected against cartilage degradation Inhibited Mmp-1a-, Mmp-3-, Mmp-13-, and Adamts-5-positive cells Effects in cartilage/OA-induced mice: Improved cartilage surface in a dose-dependent manner and decreased OARSI score Inhibited *Mmp1a, Mmp3, Mmp13*, and *Adamts5* mRNA expression and positive cells	[235]
Sclareol	*Salvia sclarea* Diterpene	Effects in IL-1β-induced chondrocytes: Chondroprotective properties and no adverse effects on cell viability with concentrations of 1–10 μg/mL Inhibited *Mmp1, Mmp3, Mmp13, Cox2* and *Nos2* gene and protein expression Suppressed Mmp1, Cox2 and Nos2 protein level Inhibited NO and Pge2 production Upregulated *Timp1* gene and protein expression Effects in cartilage/OA-induced rabbit: Decreased *Mmp1, Mmp3, Mmp13, Cox2* and *Nos2* and increased *Timp1* gene expression Ameliorated cartilage degradation by intra-articular injection and reduced Mankin score	[326]
Sesamin	*Sesamun indicum* sesame seed oil lignan	Effects in porcine cartilage explants: Inhibited degradation of PG cultures treated with IL-1β Inhibition of IL-1β/OSM-induced collagen degradation and hydroxyproline release Effects in cartilage/papain-induced OA rat Inhibited cartilage degradation and OA progression Increased PG and Col2a1 deposition in a dose-dependent manner	[237]
Shikonin	Dried roots of *Lithospermum* *erythronrhizon* Naphthoquinone (phenols)	Effects in blood samples/OA tissue/OA-induced rat Inhibited inflammation and inhibited Il-1β, Tnf-α and Nos2 in blood Suppressed Nf-κb pathway protein expression Decreased Cox-2 protein expression and Casp-3 activity Upregulated phosphorylated Akt protein level	[327]
		Effects in IL-1β-induced rabbit chondrocytes: Anti-inflammatory and chondro-protective properties Inhibited *Mmp1, Mmp3* and *Mmp13* gene and protein expression Increased *timp1* gene and protein expression Suppressed Nf-κb p65 activation Suppressed Iκbα degradation Effects in cartilage/OA-induced rabbit: Decreased cartilage damage by intra-articular injection treatment Suppressed *Mmp1, Mmp3* and *Mmp13* gene Enhanced *Timp1* gene expression	[328]
		Effects in IL-1β-induced rat chondrocytes: Reduced the cytotoxicity induced by IL-1β Inhibited chondrocyte apoptosis by enhancing Pi3k/Akt signalling pathway Suppressed Casp-3 activation and reduced cytochrome c release Increased Bcl-2 and decreased Bax expression Inhibited *Mmp13* mRNA and protein expression Increased *Timp1* mRNA and protein expression	[329]
Sinomenine	*Sinomenium acutum* Alkaloids	Effects in IL-1β-treated rabbit cartilage explants: Showed chondroprotective effects Inhibited PG degradation Suppressed *Mmp3* gene and protein expression Upregulated *Timp1* mRNA and protein expression in a dose-dependent manner Effects in IL-1β-induced chondrocytes: Reduced DNA fragmentation Inhibited Casp-3 activity and apoptotic chondrocytes in a dose-dependent manner	[330]
		Effects in IL-1β-induced mice chondrocytes: Inhibited inflammatory response and ECM degradation in a dose-dependent manner Decreased Mmp-3, Mmp-13 and Adamts-5 levels Upregulated Col2a1 and Acan synthesis Inhibited NO, Pge2, Nos2, Cox-2, Il-6 and Tnf-α protein levels Protected against OA progression via the activation of Nrf2/Ho-1 the signalling pathway and the inhibition of p-Nf-κb p65 nuclear translocation and activation, and inhibited Iκbα degradation Effects in cartilage/OA-induced mouse: Reduced OARSI scores and inhibited cartilage degradation	[331]
Sulforaphane	*Brassica oleracea italica* cruciferous vegetables (abundant in broccoli) Isothiocyanate	Effects in IL-1/OSM-induced bovine nasal cartilage explant/OA induced murine Showed chondroprotective effects Inhibited GAG and hydroxyproline release Inhibited cartilage destruction	[241]
SFX-01^®^, a stable synthetic form of sulforaphane	Synthetic sulforaphane- alpha-cyclodextrin inclusion complex	Effects in STR/Ort OA mice: Led to greater symmetry in gait Improved bone microarchitecture Reduced osteoclast number and bone resorption Enhanced trabecular bone mass in the metaphyseal compartment Enhanced cortical bone mass Decreased Ctx-I protein levels in serum Increased procollagen type I NH2-terminal propeptide protein level in serum	[332]
Sulforaphane– microsphere system	Sulforaphane-Poly (D, L-lactic-co-glycolic) acid (PLGA) microspheres	Effects in cartilage/OA-induced rat: Decreased cartilage degradation and OA progression by intra-articular injection system Decreased fibrillation, PG loss and OARSI score Reduced synovial inflammation	[243]
Terpenoid compounds (tuberatolide B, loliolide, sargachromenol, sargachromanol D, sargachromanol G, sargaquinoic acid, sargahydroquinoic acid, isoketochabrolic acid/IKCA, isonahocol E3 and fucosterol) Phlorotannins Eicosapentaenoic acid EPA	*Sargassum seaweed* (Terpenoids) Polyphenols Fatty acid	Effects in IL-1β-induced rat chondrocytes: Demonstrated antioxidant activity Inhibited *Nos2* and *Cox2* mRNA and protein expression Decreased NO, Pge2 production	[245]
Triterperne concentrates (lupeol, α-amyrin, β-amyrin, butyrospermol)	*Vitellaria paradoxa* nut triterpenoids	Effects in plasma/knee cartilage/OA-induced obese rat: Reduced oxidative stress and suppressed proinflammatory cytokines Enhanced enzymatic antioxidant activities Reduced total cholesterol and increased high-density lipoprotein-cholesterol in blood plasma sample Decreased Tnf-α, Il-1β, and Il-6 levels Reduced malondialdehyde (lipid peroxidation) level and NO release in plasma Attenuated cartilage damage and suppressed OA development Reduced knee swelling, weight-bearing and pain	[333]
Wogonin	The root extract of: *Scutellaria baicalensis* Flavone	Effects in IL-1β-induced rabbit chondrocytes: Showed chondroprotective effects Inhibited *Mmp3, Mmp1*, *Mmp13*, and *Adamts4* and restored *Col2a1* gene expression Inhibited Mmp3 protein synthesis and its caseinolytic activity Effects in IL-1β-induced cartilage/OA-induced rats: Inhibited Mmp3 production via intraarticular injection into the knee joint (dose 50 or 100 μM)	[334]
		Effects in cartilage/OA-induced mice: Demonstrated efficacy and safety as a transdermal cream treatment Inhibited OA progression, and reduced OARSI and Mankin scores Increased running wheel activity and decreased pain perception Decreased biomarkers associated with cartilage degradation Inhibited Tgf-β1, Htra1, Mmp-13 and Nf-κb protein expression	[335]

**Table 3 pharmaceuticals-17-01148-t003:** Bioactive compounds as epigenetic modulators for the management, treatment, or prevention of OA in humans.

Bioactive Compounds	Sources/Classes	Effects of Bioactive Compounds	Ref.
Baicalin	(*Scutellaria baicalensis* *Georgi*) Mainly extracted from dry root Flavone glycoside (Flavonoid)	High lncRNA HOTAIR expression levels inhibited in OA chondrocytes Reduction in p-PI3K and p-AKT protein levels Increase in PTEN, APN and ADIPOR1 protein levels	[357]
		Effects in IL-1β-induced OA chondrocytes: Protected against ECM degradation and apoptosis Restored autophagy activity via the upregulation of miR-766-3p BAX and cleaved-caspase-3 expression suppression Promoted BCL-2 protein expression and increased GAG content	[358]
		Effects in IL-1β-induced OA chondrocytes: Protected against inflammatory injury Deactivated NF-κB signalling pathway by downregulation of miR-126 on IL-1β-stimulated cells IL-6, IL-8 and TNF-α downregulation and decreased cell apoptosis	[359]
Cryptotanshinone	(*Salvia miltiorrhiza* *Bunge*) Extracted from the root of the plant Diterpene quinones	Effects in chondrocytes: Increased miR-106a-5p and *PAX5* expression miR-106a-5p was positively associated with *PAX5* and negatively correlated with *GLIS3* expression Effects on tissues: PAX5/miR-106a-5p/GLIS3 regulation protects against cartilage degradation	[360]
Epigallocatechin-3-gallate	*Camellia sinensis* Green tea Flavan-3-ols (flavanols)	Effects in OA patients’ cartilage tissues and IL-1β-stimulated chondrocytes: Increases viability and decreases miR-29b-3p, MMP-12 and IL-6 levels in cells MiR-29b-3p mimics reversed the effects above 50 μM EGCG, and these effects were revoked by PTEN overexpression	[361]
		Effects in IL-1β-induced OA chondrocytes: Inhibited inflammatory response via modulation of miRNAs expressions Inhibited *ADAMTS5* gene expression via upregulation of miR-140-3p Decreased let-7e-5p, miR-103a-3p, miR-151a-5p, miR-195-5p, miR-222-3p, miR-23a-3p, miR-23b-3p, miR-26a-5p, miR-27a-3p, miR-29b-3p, miR-3195, miR-3651, miR-4281, miR-4459, miR-4516, miR-762, and miR-125b-5p Upregulated let-7 family, miR-140-3p, miR-193a-3p, miR-199a-3p, miR-27b-3p, miR-29a-3p, miR-320b, miR-34a-5p, miR-3960, miR-4284, miR-4454, miR-497-5p, miR-5100, and miR-100-5p	[362]
		Effects in OA chondrocytes: Inhibited inflammatory response via miRNAs expression modulation miR-199a-3p upregulation inhibited COX2 expression and PGE2 production	[363]
Fisetin	Persimmons, mangos, grapes, apples, peaches, strawberries, peaches, onions, tomatoes, and cucumbers *Acacia greggii*, *Acacia berlandieri*, *Butea frondosa*, *Gleditsia triacanthos*, *Quebracho colorado* Flavonol	Effects in IL-1β-induced OA chondrocytes: Showed anti-inflammatory effects through activating SIRT-1 Inhibited the degradation of *SOX9*, *ACAN* and *COL2A1* mRNA and protein expression Decreased NO, PGE2, IL-6, TNF-α production Inhibited *NOS2, COX2, MMP3, MMP13* and *ADAMTS5* expression at the mRNA and protein levels	[364]
Hydroxytyrosol (HT)	*Olea europea* L. fruits and leaves Extra virgin olive oil Secoiridoid derivative	Effects in C-28/I2 and primary OA chondrocytes: Showed chondroprotective and antioxidant effects Protected from DNA damage and cell death induced by oxidative stress Increased *P62* mRNA transcription and autophagy activation by SIRT1 pathways	[365]
		Effects in OA chondrocytes: Oxidative stress and DNA damage reduction Prevented the increase in cell death and caspases activation Decreased expression of pro-inflammatory genes (*COX2, NOS2*) and of genes involved in chondrocyte terminal differentiation (*RUNX2, MMP13* and *VEGF*) Increased *SIRT1* mRNA expression in GROa-stimulated micromasses	[366]
		Effects in C-28/I2 and OA chondrocytes: Protected against oxidative stress and modulated through epigenetic mechanism Reduced miR-9 levels (involved in oxidative stress and influenced OA-related gene expression) by enhancing SIRT-1 Reduced *MMP13, VEGF* and *RUNX2* genes	[367]
		Effects in C-28/I2 chondrocytes: miR-9 promoters’ demethylation by *SIRT1* silencing miR-9 promoters’ hypomethylation in H_2_O_2_-treated cells and hypermethylation in cells treated with HT alone or together with H_2_O_2_ under oxidative stress conditions	[368]
Oleanolic acid	*Ligustri lucidi* extracted from fructus pentacyclic triterpenoid	Showed SIRT3 anti-inflammatory effect, preventing IL-1β-induced FLS dysfunction *in vitro* SIRT3 activation inhibited synovial inflammation by NF-κB signal pathway suppression in FLS	[369]
		Effects in IL-1β-induced chondrocytes: Alleviated chondrocytes’ growth inhibition and the cell membrane and DNA damage Protective effects induced by activating miR-148-3p-mediated FGF2 Showed antiapoptotic effect by the inhibition of FGF2	[370]
Quercetin	(*Achyranthes bidentata*) (*Ageratum conyzoides*) flowers, leaves, and fruits of plants such as *Chrysanthemum* *psyllium, Eleutherococcus senticosus, Juglans regia* L. onions, apples, broccoli, berry crops, grapes, dark cherries, and green vegetables Flavonol (Flavonoid)	Role of BMSC-derived exosomes both in vitro and in vivo (OA patients) Conditioned medium of quercetin-treated BMSCs was able to revert IL-1β effects in chondrocytes (decreased MMP13 and ADAMT5, and increased COL2A1 expression) OA progression inhibition through miR-124-3p upregulation	[371]
Resveratrol	Root extracts of the weed: *Polylygonum cuspidatum* *Vitis vinifera* Red grapes, blueberries cranberries, peanuts Stilbenes (polyphenols)	*In vitro* studies in IL-1β-treated chondrocytes: Resveratrol increased SIRT1 expression and FoxO1 phosphorylation, promoting the expression of cholesterol efflux factor liver X receptor alpha, and inhibiting the expression of cholesterol synthesis-associated factor sterol-regulatory element binding proteins 2, reducing cholesterol accumulation *In vivo* experiments showed that RES can alleviate cholesterol build-up and pathological changes in OA cartilage via the SIRT1/FoxO1 pathway	[372]
		Bioinformatics methods allowed us to identify 1016 differentially expressed lncRNAs (493 downregulated) between control and resveratrol-treated chondrocytes	[373]
		Effects in OA chondrocytes: Increased *SIRT1* mRNA and protein expression SIRT-1 regulated apoptosis and ECM degradation via the WNT/β-catenin signalling pathway Decreased BAX, proCASP-3 and proCASP-9, MMP-1, MMP-3, MMP-13, WNT3A, WNT5A, WNT7A, and CTNNB1 protein expression	[374]
		Effects in IL-1β-induced chondrocytes: Prevented OA progression by increase in SIRT1 and silencing NF-κB p65 and HIF-2α Decreased *NOS2* and *MMP13* and reestablished *COL2A1* and *ACAN* gene expression	[49]
		Effects in OA osteoblasts/subchondral bone tissue: Reduced ALP activity at a high dose Upregulated SIRT-1 activity and reduced the expression of leptin Increased the mineralization Increased the phosphorylation of ERK1/2 and WNT/β-catenin signalling	[375]

**Table 4 pharmaceuticals-17-01148-t004:** Bioactive compounds as epigenetic modulators for the management, treatment, or prevention of OA in animals.

Bioactive Compounds	Sources/Classes	Effects of Bioactive Compounds	Ref.
Cryptotanshinone	(*Salvia miltiorrhiza* *Bunge*) Extracted from the root of the plant Diterpene quinones	Effects in OA mouse model: Affects chondrocyte apoptosis by regulating miR-574-5p expression and then interfering with YAF2 Regulates miR-574-5p promoter methylation	[376]
Curcuminoids: Curcumin Demethoxycurcumin, Bisdemethoxycurcumin	(*Curcuma longa*) (*Curcuma domestica*) Turmeric rhizome Diarylheptanoids (Phenolic compounds)	Effects in knee OA rat model: Protective effect against quadriceps femoris atrophy and improves knee OA ROS-induced autophagy decreases via the SIRT3-SOD2 pathway	[377]
		Effects in TBHP-treated rat chondrocytes: Protected from oxidative stress-induced apoptosis Suppressed ER stress biomarkers Perk-Eif2a-Atf4-Chop pathway via activation of the mRNA and *Sirt1* protein expression Increased *Col2a1* and *Bcl2* gene expression and downregulated cleaved-Casp-3 and cleaved-Parp (proapoptotic proteins) levels Effects in cartilage/OA-induced rat: Demonstrated therapeutic efficacy (treatment: 50 mg/Kg and 150 mg/Kg once daily for 8 weeks by intraperitoneal injection) Attenuated knee joint degradation and inhibited OA progression Reduced cleaved-Casp-3 and Chop levels Activated Sirt1 expression and decreased chondrocyte apoptosis and ER stress Ameliorated chondrocytes and PG loss Decreased OARSI score in a dose-dependent manner	[378]
		Effects in IL-1β-induced primary chondrocytes/OA-induced mice: Attenuated OA progression and decreased apoptosis by exosomes derived from curcumin-treated mesenchymal stem cells Upregulated miR-143 and miR-124 expression by hypomethylation of their promoters Inhibited *Nfkb, Rock1* and *Tlr9* mRNA and protein expression	[379]
Fisetin	Persimmons, mangos, grapes, apples, peaches, strawberries, peaches, onions, tomatoes, and cucumbers *Acacia greggii*, *Acacia berlandieri*, *Butea frondosa*, *Gleditsia triacanthos*, *Quebracho colorado* Flavonol	Effects on DMM rats and IL-1β-treated chondrocytes: FST can activate SIRT6 Positive effects against inflammation, ECM degradation, apoptosis, and senescence in IL-1β-stimulated chondrocytes In chondrocytes, FST reduces injury-induced aging-related phenotype changes via SIRT6 targeting	[380]
		Effects in cartilage/subchondral bone/synovium/ OA-induced mice models Exhibited less cartilage destruction and attenuated OA progression Decreased OARSI score Reduced subchondral bone plate thickness Alleviated synovitis	[364]
Hydroxytyrosol (HT)	*Olea europea* L. fruits and leaves Extra virgin oil Secoiridoid derivative	Effects in TNF-α-induced rat chondrocytes: Showed anti-inflammatory activity Inhibited Il-1β, Il-6 and Mcp-1 proteins by upregulating *Sirt6* mRNA and protein levels Promoted autophagy process through *Sirt6*	[381]
Quercetin	(*Achyranthes bidentata*) (*Ageratum conyzoides*) flowers, leaves, and fruits of plants such as *Chrysanthemum psyllium*, *Eleutherococcus senticosus, Juglans regia* L. Onions, apples, broccoli, berry crops, grapes, dark cherries, and green vegetables Flavonol (Flavonoid)	Inhibited the expression of IL-1β-induced MMP-3, MMP13, iNOS and COX-2, and promoted COL type II expression *in vitro*. This effect is mediated by SIRT1/Nrf-2/HO-1 activation and ferroptosis inhibition	[382]
		In an ACLT-OA rat model, QUE treatment improved articular function (cartilage damage, joint pain, and subchondral bone remodelling). QUE also reduced serum IL-1β, TNF-α, MMP3, CTX-II, and COMP, thereby slowing the progression of OA	[383]
		Effects in chondrocytes/OA-induced rat: Chondroprotective and antioxidant properties Inhibited oxidative and endoplasmic reticulum stress, and chondrocyte apoptosis by activating Sirt-1 and Ampk signalling pathway Downregulated Chop, Grp78, P-perk, P-ire1α, Atf6 (ERstress biomarkers), cleaved-Casp-3 and cleaved-Parp (apoptosis biomarkers) levels Upregulated Bcl-2 protein expression levels Attenuated cartilage degradation of knee joint (dose: intraperitoneal injection of 50–100 mg/Kg once daily, 12 weeks)	[384]
		Effects in rat OA chondrocytes: Upregulated Ampk/Sirt-1 signalling pathway Effects on cartilage/blood/OA-induced rat: Inhibited inflammation, mitochondrial dysfunction and ROS (100 mg/Kg oral treatment/daily, 7 days) Increased ATP, GSH and GPx levels Inhibited nitrite, Mmp-3 and Mmp-13 levels in blood samples	
Resveratrol	Root extracts of the weed: *Polylygonum cuspidatum* *Vitis vinifera* Red grapes, blueberries cranberries, peanuts Stilbenes (polyphenols)	ECM metabolism, autophagy, and apoptosis regulation of OA chondrocytes via SIRT1/FOXO1 pathway to improve IL-1β-induced chondrocyte damage	[385]
		Effects in OA cartilage/OA-induced mice: Prevented OA cartilage destruction and improved cartilage structure (dose: 100 µg) by intraarticular injection Increased Sirt-1 expression and reduced Nf-κb p65 and Hif-2α Reduced subchondral bone plate thickness and prevented calcified cartilage damage Decreased Nos2 and Mmp-13 and inhibited Col2a1 degradation and PG loss	[49]
		Effects in chondrocytes/cartilage/OA-induced mice: Promoted chondroprotective effects by intra-articular injection chondrocyte and increased the growth rate of chondrocyte Reduction in Il-6, Mmp-13 and Casp-3 protein expression levels Increased miR-9 expression levels Decreased Malat1 and Nfkb1 gene and protein expression Malat1 and Nfκb1 were identified as potential target genes of miR-9	[386]
		Effects in IL-1β-induced rat chondrocytes: Exerted anti-inflammatory properties and inhibited Nf-κb signalling pathway by activating Sirt-1 Suppressed *Nos2* expression and NO production Decreased DNA-binding activity of p65 by upregulation of Sirt-1 Inhibited Lys310-acetylated p65 accumulation in the nucleus	[387]
Saikosaponin D	*Radix bupleri* Triterpene saponin	*In vivo*, SSD decreased cartilage damage and inflammatory factors and induced autophagy in OA mice MiR-199-3p expression was downregulated and transcription factor-4 expression was upregulated in cartilage *In vitro* experiments showed that SSD decreased the inflammation and induced autophagy in OA chondrocytes MiR-199-3p downregulation attenuated the SSD effect on OA chondrocytes	[388]
Sinomenine	*Sinomenium acutum* Alkaloids	Effects on cartilage/OA mice: Inhibited cartilage damage by miR-223-3p upregulation via inactivation of the inflammasome signalling Nlrp3 was a direct target of miR-223-3p Blocked inflammatory markers (Tnf-α, Il-1β, Il-6, and Il-18) Effects in chondrocytes: MiR-223-3p overexpression inhibited both IL-1β-induced apoptosis and Il-1β and Il-18 levels	[389]
TXC compound: Paeoniflorin Ferulic acid Isofraxidin Rosmarinic acid	Dried roots of: (*Paeonia lactiflora Pall, Morindae officinalis* *Ligusticum wallichii* *Sarcandra glabra*) Monoterpene glycosides Hydroxycinnamic acid Coumarin Hydroxycinnamic acid	Effects in knee OA cartilage/subchondral bone/OA-induced rats: Showed therapeutic effects in cartilage protection and subchondral bone remodelling Downregulated *Mmp9, Adamts5, Col5a1, Col1a1, Mmp3, Mmp13,* and *Postn* gene and protein expression Effects in LPS-exposed rat chondrocytes: Decreased Il-1β, Il-6, Tnf-α, Mmp-9 and p38 MAPK pathway in LPS-exposed chondrocytes Increased miR-27b, miR-140, and miR-92a-3p and decreased miR-34a expression Suppressed *Adamts4, Adamts5, Mmp3,* and *Mmp13* mRNA and protein expression	[390]

## Data Availability

Data is contained within the article.

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
