# Peer review of "Nutritional Epigenomics: Bioactive Dietary Compounds in the Epigenetic Regulation of Osteoarthritis"

_pharmaceuticals, 2024, doi:10.3390/ph17091148_

Round 1

Reviewer 1 Report

Comments and Suggestions for Authors

Manuscript Title: Nutritional Epigenomics: Bioactive Dietary Compounds in the Epigenetic Regulation of Osteoarthritis

Author have summarised the chondroprotective properties of bioactive compounds and nutraceuticals for the management, treatment, or prevention of OA in both human and animal studies.

It can be accepted after addressing some concerns which are below mentioned.

1.       Similarity as per report attached is too high, 36%. Please reduce it to 20-25%.

2.       Line 22: Avoid citation in abstract. Improve the abstract too with more information and avoiding repetitive content.  

3.       Abbreviations should start from the introduction, not from the abstract. Kindly correct it accordingly.

4.       The review paper's major content is in tabular form. Need to explain the content in text and correlate with each other.

5.       The author should include a discussion of bioactive compounds in the text too for better understanding.

6. Table 3 in the section of sources and subclass is written only for Ref 357. What about other points, no subclass is written for the rest in the table.

7.       Check formatting of Table 4.

8.       In table 4 again some of the subclass is missing.

9.       Author should add patents related to Bioactive compounds as epigenetic modulators for osteoarthritis.

10.   Author should add future perspectives on Bioactive Dietary Compounds in osteoarthritis.

11.   Author should also add the list of abbreviations at the end of the manuscript.

12.   Author should add one 600 DPI figure to Point no 4. (Bioactive Compounds: Health-Protective Benefits)

13.   Reivew lack the original figures. Please add one or two figures.

14.   Inclusion of limitation and toxicity study data would make this review more useful. 

Author Response

We would like to thank all three reviewers for their constructive criticisms that have improved our manuscript considerably. We have addressed their comments as follow (changes are highlighted in green):

REVIEWER 1

Comment 1: Similarity as per report attached is too high, 36%. Please reduce it to 20-25%.

Response: We thank the reviewer for this comment. We have thoroughly checked the manuscript and reduced the similarity to 23% (www.turnitin.com). Nevertheless, we founded that a high proportion of similarity was due to the list of references. Changes along the text are highlighted in yellow.

Comment 2: Line 22: Avoid citation in abstract. Improve the abstract too with more information and avoiding repetitive content.  

Response: We thank the reviewer for pointing out this typo, now reference 1 is at its right location (line 34). Also, we have revised the abstract and modified accordingly. To avoid repetitive content, we have deleted two sentences: “A healthy diet can improve the quality of life and, alleviate the progression and symptomatology of many complex diseases such as osteoarthritis” and, “Some of them have been considered as natural epigenetic modulators that can modify the activity of various epigenetic factors and, alter the expression of genes related to inflammation and cartilage destruction”. Furthermore, we have added some more information exemplified in this sentence: “to determine their potential value for future clinical applications in osteoarthritic patients goes along with genomic and nutritional environment in order to personalise food and nutrition together with disease prevention”.

Comment 3: Abbreviations should start from the introduction, not from the abstract. Kindly correct it accordingly.

Response: We thank the reviewer for pointing out this typo, we have removed the abbreviations from the abstract.

Comment 4: The review paper's major content is in tabular form. Need to explain the content in text and correlate with each other.

Response: We thank the reviewer for this insightful comment. We have added some statements referring to the main findings reflected in the tables.

For tables 1 and 2 (lines 310-318): In OA, most studied bioactive compounds are curcuminoids [164-184,274-279], epigallocatechin-3-O-gallate [187-193], hydroxytyrosol [208-210,296], icariin [298-302], oleuropein [223-224], resveratrol [228-234,321-324] and, sulphuronate [238-242,241-243]. The commonest effects founded in vitro are related to decreased inflammatory and cartilage degradation markers, like MMPs, NO, PGE2 or ROS. On the other hand, in vivo effects observed in OA-induced animal models are critically linked to the reduction of symptoms at joint level (cartilage, synovium and subchondral bone). Finally, case studies were carried out in humans, showing alleviated pain and enhanced quality of life among other symptoms.

For tables 3 and 4 (lines 393-398): Few, but insightful studies have shown epigenetic effects for bioactive compounds in OA. The majority of studies are focused on curcuminoids [377-379], epigallocatechin-3-O-gallate [361-363], hydroxytyrosol [365-368,381], oleanoic acid [369-370] and, resveratrol [372-375,385-387]. By far, the most studied epigenetic mechanism are miRNAs, generally linked to the regulation of inflammatory and cartilage degradation markers. Sirtuins are also well explored in the context of OA.

Comment 5: The author should include a discussion of bioactive compounds in the text too for better understanding.

Response: We thank the reviewer for this comment. In order to complete the discussion of bioactive compounds we have added the following paragraph (lines 319-323): “Regarding bioactive compounds applications there are important considerations to take into account: i) it will be crucial to increase their stability and bioavailability, especially for clinical applications; ii) deep understanding of the underlying molecular mechanisms to increase their bioactivity; and iii) investigate their long-term toxicity and possible side effects”.

Comment 6: Table 3 in the section of sources and subclass is written only for Ref 357. What about other points, no subclass is written for the rest in the table.

Response: We thank the reviewer for this comment. We have double-checked all the compounds and added sources and subclasses for fisetin for both Table 3 and Table 4. For the example pointed out by Reviewer 1: References 357-359, all belong to the same compound baicalin, for that reason it only appears once.

Comment 7: Check formatting of Table 4.

Response: We thank the reviewer for pointing out this format issue, we have tried to homogenise the format for all 4 tables. This is issue should be solved once the manuscript will be converted to a pdf file, tables format change slightly from one computer to another.

Comment 8: In table 4 again some of the subclass is missing.

Response: Addressed in comment 6.

Comment 9: Author should add patents related to Bioactive compounds as epigenetic modulators for osteoarthritis.

Response: We thank the reviewer for this comment, however, after checking two official websites for patents (Espacenet and Patentscope), so far there are not patents for bioactive compounds as epigenetic modulators for osteoarthritis.

Comment 10: Author should add future perspectives on Bioactive Dietary Compounds in osteoarthritis.

Response: We thank the reviewer for this thoughtful comment. We have added the following paragraph to the conclusion section (lines 433-439): “Future perspectives of bioactive dietary compounds in OA are mainly preventive more than therapeutic. Mostly because the effects of these natural products probably are very small during short periods of time; however, they could be effective when consumed continuously as part of the diet. This indeed could be crucial for a disease like OA, where prevention before symptoms appear is key to stop the progression of the disease”.

Comment 11: Author should also add the list of abbreviations at the end of the manuscript.

Response: We thank the reviewer for this comment. We have added a list of the most used abbreviations at the end of the manuscript and, we have checked thoroughly the document to modify accordingly.

Comment 12: Author should add one 600 DPI figure to Point no 4. (Bioactive Compounds: Health-Protective Benefits)

Response: Following this reviewer’s recommendation, we have elaborated a figure (now Figure 1) showing a schematic representation of the main food bioactive compounds classification. We added two plant-based representative foods, as well as sources and an illustrative chemical structure example.

Comment 13: Review lack the original figures. Please add one or two figures.

Response: We do agree with the reviewer about this comment. However, due to the considerable manuscript length, we have added just one figure as explained in comment 12.

Comment 14: Inclusion of limitation and toxicity study data would make this review more useful. 

Response: We thank the reviewer for this suggestion. These important topics are now mentioned in the paragraph (lines 319-323): “Regarding bioactive compounds applications there are important considerations to take into account: i) it will be crucial to increase their stability and bioavailability, especially for clinical applications; ii) deep understanding of the underlying molecular mechanisms to increase their bioactivity; and iii) investigate their long-term toxicity and possible side effects”.

Reviewer 2 Report

Comments and Suggestions for Authors

The manuscript “Nutritional Epigenomics: Bioactive Dietary Compounds in the Epigenetic Regulation of Osteoarthritis” aims to explore the importance of bioactive compounds as epigenetic modulators in the prevention and treatment of Osteoarthritis. The article is well organized and easy to follow. Please find below comments which may be useful.

Comments:

1.      Elaborate the discussion on section 4 bioactive compounds with case studies.

2.      Revise and add more information to the conclusion section.

I would support the publication of this manuscript with the minor revision.

Author Response

We would like to thank all three reviewers for their constructive criticisms that have improved our manuscript considerably. We have addressed their comments as follow (changes are highlighted in green):

REVIEWER 2

Comment 1: Elaborate the discussion on section 4 bioactive compounds with case studies.

Response: We thank the reviewer his/her constructive comments. Following also Reviewer 1 suggestion, we have added some content to text to correlate with tables. Specifically, we have mentioned case studies as suggested by the reviewer:

(Lines 310-328)

In OA, most studied bioactive compounds are curcuminoids [164-184,274-279], epigallocatechin-3-O-gallate [187-193], hydroxytyrosol [208-210,296], icariin [298-302], oleuropein [223-224], resveratrol [228-234,321-324] and, sulphuronate [238-242,241-243]. The commonest effects founded in vitro are related to decreased inflammatory and cartilage degradation markers, like MMPs, NO, PGE2 or ROS. On the other hand, in vivo effects observed in OA-induced animal models are critically linked to the reduction of symptoms at joint level (cartilage, synovium and subchondral bone). Finally, case studies were carried out in humans, showing alleviated pain and enhanced quality of life among other symptoms. Several case studies showed interesting results compared to the conventional analgesic therapy taken by OA patients, especially curcuminoids. It has been proven as they can be as efficacious as ibuprofen [168-169], showed potential beneficial effects as adjuvant therapy with diclofenac [170] and meloxican [229], alternative therapy for intolerant to diclofenac side effects [171], reduced use of NSAIDs and gastrointestinal complications [178] and, lower adverse effects than diacerhein [205-206].

Comment 2: Revise and add more information to the conclusion section.

Response: We thank the reviewer for this suggestion. Accordingly, we have added this paragraph to the conclusion section in order to add more information (lines 433-439): “Future perspectives of bioactive dietary compounds in OA are mainly preventive more than therapeutic. Mostly because the effects of these natural products probably are very small during short periods of time; however, they could be effective when consumed continuously as part of the diet. This indeed could be crucial for a disease like OA, where prevention before symptoms appear is key to stop the progression of the disease. Finally, it will be critical to identify biomarkers to test the efficacy of bioactive compounds at both inter-individual and population levels”.

Reviewer 3 Report

Comments and Suggestions for Authors

I would like to express my appreciation for the well-structured and detailed article. The authors have done an excellent job of linking the complex concepts of epigenomics and nutrition to a prevalent and debilitating clinical condition such as osteoarthritis (OA). The ability to clearly and comprehensibly explain how natural epigenetic modulators can influence gene expression and potentially alter the course of OA is remarkable. Furthermore, the article is enriched by a solid review of the literature, demonstrating a deep understanding of this evolving field. While reading your commendable study, I found many points of agreement. In this regard, the connection between nutrition and epigenetics is well articulated, and the explanation of the impact that natural epigenetic modulators can have on gene expression is compelling. Moreover, the article highlights how a diet rich in bioactive compounds can influence epigenetic mechanisms related to inflammation and cartilage degeneration, which are central aspects in the pathogenesis of OA. This is a promising approach that deserves further research and clinical consideration. The emphasis on tailoring dietary recommendations based on individual epigenetic profiles is a strong point of the article. The authors acknowledge the importance of developing personalized nutritional strategies, which could lead to more effective and targeted outcomes. This approach aligns with the growing trend toward precision medicine, which seeks to tailor treatments to the individual characteristics of the patient. The article rightly emphasizes the potential of nutritional epigenomics not only in the management but also in the prevention of OA. This is particularly relevant given that OA is a chronic disease for which preventive options are currently limited. The idea of using nutrition to modulate epigenetic pathways early on is intriguing and could have a significant impact on public health. While the article recognizes the need to integrate natural epigenetic modulators with conventional pharmacological therapies, a deeper exploration of potential interactions between these compounds and commonly used drugs for OA would be useful. Considering that many patients with OA take anti-inflammatory and analgesic medications, it is crucial to understand how epigenetic modulators might affect the efficacy or safety of these therapies. The idea of monitoring epigenetic modifications as biomarkers of disease progression and treatment response is excellent, but further details on specific biomarkers that could be used would be needed. For instance, which genes or epigenetic pathways should be monitored to assess the effectiveness of nutritional modulators? A more detailed analysis of these aspects would make the article even more comprehensive. Although the article does a good job of summarizing the current evidence, it would be useful to include a section that critically examines the strength and limitations of the available evidence. For example, how many clinical studies have actually tested natural epigenetic modulators in OA? What are the main findings, and what are the research gaps? This would help to better understand the scientific context in which the authors' proposal is situated. The article rightly raises concerns about the accessibility of diets rich in epigenetic modulators and the ethical implications of personalized therapies. It would be helpful to further explore how health policies could address these challenges, highlighting specific measures that could be adopted to ensure that nutritional epigenomics-based therapies are accessible to all patients, regardless of their socioeconomic status.

In conclusion, the article represents a significant contribution to the emerging field of nutritional epigenomics applied to osteoarthritis. The clear presentation of the therapeutic and preventive potential of natural epigenetic modulators is an invitation to further research and clinical consideration of these approaches. However, as in any emerging field, greater depth is needed in certain key areas, including interaction with pharmacological therapies, identification of specific biomarkers, and critical evaluation of the available evidence.

Author Response

We would like to thank all three reviewers for their constructive criticisms that have improved our manuscript considerably. We have addressed their comments as follow (changes are highlighted in green):

REVIEWER 3

We really thank the reviewer his/her constructive comments. Following all three reviewers some statements have been added to the manuscript in order to clarify most of the comments raised by this reviewer:

  • Interaction with pharmacological therapies (lines 318-323]:

Several case studies showed interesting results compared to the conventional analgesic therapy taken by OA patients, especially curcuminoids. It has been proven as they can be as efficacious as ibuprofen [168-169], showed potential beneficial effects as adjuvant therapy with diclofenac [170] and meloxican [229], alternative therapy for intolerant to diclofenac side effects [171], reduced use of NSAIDs and gastrointestinal complications [178] and, lower adverse effects than diacerhein [205-206].

  • Which genes or epigenetic pathways should be monitored to assess the effectiveness of nutritional modulators? (lines 393-398):

Few, but insightful studies have shown epigenetic effects for bioactive compounds in OA. The majority of studies are focused on curcuminoids [377-379], epigallocatechin-3-O-gallate [361-363], hydroxytyrosol [365-368,381], oleanoic acid [369-370] and, resveratrol [372-375,385-387]. By far, the most studied epigenetic mechanism are miRNAs, generally linked to the regulation of inflammatory and cartilage degradation markers. Sirtuins are also well explored in the context of OA.

  • Bioactive compounds applications (lines 324-328]

Regarding bioactive compounds applications there are important considerations to take into account: i) it will be crucial to increase their stability and bioavailability, especially for clinical applications; ii) deep understanding of the underlying molecular mechanisms to increase their bioactivity; and iii) investigate their long-term toxicity and possible side effects.

  • Future perspectives (lines 433-439):

Future perspectives of bioactive dietary compounds in OA are mainly preventive more than therapeutic. Mostly because the effects of these natural products probably are very small during short periods of time; however, they could be effective when consumed continuously as part of the diet. This indeed could be crucial for a disease like OA, where prevention before symptoms appear is key to stop the progression of the disease. Finally, it will be critical to identify biomarkers to test the efficacy of bioactive compounds at both inter-individual and population levels.

  • Personalised therapies (lines 22-25):

Determine their potential value for future clinical applications in osteoarthritic patients goes along with genomic and nutritional environment in order to personalise food and nutrition together with disease prevention.